# BCUB - A large sample ungauged basin attribute dataset for British Columbia, Canada.

Daniel Kovacek[1] and Steven Weijs[1]

[1]Department of Civil Engineering, University of British Columbia, 2002 - 6250 Applied Science Lane, Vancouver, V6T 1Z4, BC, Canada

**Correspondence:** Daniel Kovacek (dkovacek@mail.ubc.ca)

**Abstract.** The British Columbia Ungauged Basin (BCUB) dataset is an open-source, extensible dataset of attributes describing terrain, soil, land cover, and climate indices of over one million ungauged catchments in British Columbia, Canada including trans-boundary regions. The attributes included in the dataset are similar to those found in the large sample hydrology literature for their association with hydrological processes. The BCUB database is intended to support water resources research and practice, namely monitoring network analysis studies, or hydrological modelling where basin characterization is used for model calibration. The dataset and the complete workflow to collect and process input data, to derive stream networks, and to delineate sub-basins and extract attributes, is available under a Creative Commons BY 4.0 license. The DOI link for the BCUB dataset is https://doi.org/10.5683/SP3/JNKZVT (Kovacek and Weijs, 2023).

## 1 Introduction

Spatial datasets available for geoscience research and practice are increasing in size, scale, resolution, and variety. Advances in the capture and processing of remote sensing data have in recent years led to open-access publication of continental and global scale geospatial datasets at high resolution (U.S. Geological Survey, 2022; Huscroft et al., 2018; Latifovic et al., 2010; Lehner et al., 2021; Thornton et al., 2021). Geospatial data availability has supported the emergence of Large Sample Hydrology (LSH) datasets, which combine streamflow and climate time series with diverse physical attributes of streamflow monitored catchments to advance hydrological research at a global scale (Addor et al., 2020). The growth of LSH datasets realizes the age of high quality, open-access geospatial data anticipated by Hrachowitz et al. (2013) following the decade of prediction in ungauged basins (PUB).

In contrast to the more recent expansion of geospatial data products, the streamflow monitoring network in Canada has contracted over the last three decades. Based on the HYDAT dataset accessed at Environment Canada's national water data archive, the number of streamflow observation locations across Canada peaked in the order of 2300 in the 1980s, and reduced to roughly 1700 in 2022 (on average per day). According to surface water monitoring density standards developed by the World Meteorological Organization (WMO) (via Coulibaly et al. (2013)), nearly 90% of Canada's terrestrial area is under-monitored, and almost 40% is classified as ungauged. In general this trend holds for the province of British Columbia (BC), where outside of a few small regions in the south it is predominantly classified as ungauged or poorly gauged (Coulibaly et al., 2013).

The streamflow data used in a wide range of research and practice today comes from monitoring networks built over many decades, highlighting the significant lag between monitoring objectives of the past and information needs of the present. Monitoring network decisions today must anticipate information needs decades into the future.

Recent deep learning (DL) approaches to regional hydrological modeling use LSH datasets to infer relationships between climate input forcings and streamflow, and model performance has been shown to improve when training incorporates static catchment attributes (Kratzert et al., 2019), though this may be due in part to providing heterogeneity to training samples to help the model differentiate between catchments in spite of uncertainty in attribute values (Li et al., 2022). DL models benefit from training datasets (streamflow monitoring networks) representing hydrologically diverse catchments, yet there is no clear consensus on how to define or evaluate such diversity (Gauch et al., 2021). A simple approach to increasing monitoring network diversity might be to add stations to the monitoring network, aligning with Beven's concept of "uniqueness of place" (Beven, 2000). Alternatively, increasing diversity could be approached by establishing a basis of comparison, such as a large sample of ungauged catchments defined by hydrologically relevant attributes.

The vast and growing amount of geospatial information available today requires considerable data assimilation effort to support specific research questions. A large, catchment-based dataset of geophysical attributes could support other disciplines that use attributes at the catchment level, for example in understanding changing water temperature and its effect on fish habitat (Daigle et al., 2017), or likewise for water quality monitoring in evaluating human-induced concentrations of toxic contaminants in fish (Scholes et al., 2016).

Water resource management decisions are typically made at the catchment level, so research and practice may be well served by datasets that are catchment-based, diverse in characteristics, and large in size and scale to reflect the scale-dependency of physical processes governing the rainfall-runoff response (Arsenault et al., 2020).

## 1.1 Motivation

The monitoring deficit of a region can be addressed by adding more stations or, under limited resources, optimal network arrangements can be approximated based on models trained on existing streamflow monitoring records, combined with information about unmonitored locations (Mishra and Coulibaly, 2010; Werstuck and Coulibaly, 2017, 2018). If large sample datasets improve predictability in ungauged locations by learning from diversity (Addor et al., 2017), a basis is needed to compare the existing monitoring network against the greater region it is intended to represent in relevant hydrological terms. The British Columbia Ungauged Basins (BCUB) (Kovacek and Weijs, 2023) is designed to be a dataset which i) uses only open access data sources that are continuous and complete over the study region, ii) is derived from the highest resolution DEM available to cover the range of catchment areas represented in large sample hydrology (monitored catchment) datasets, iii) is published under an open-source license, iv) is extensible both spatially and dimensionally to enable integration of new information as it is published, and v) is published with the full replication code based on widely used open-source libraries. Several existing datasets were reviewed for the desired qualities listed above, and for their potential to support research in network optimization, prediction in ungauged catchments, and water resources more generally.

Hydrographic datasets, unlike Large Sample Hydrology (LSH) datasets, focus on mapping hydrographic features such as rivers, lakes, and watersheds but do not include the detailed catchment-level attributes or the diversity of hydrological, climatic, and physical characteristics essential for large-scale hydrological modeling. Monitoring networks are sparse and LSH datasets do not cover the ungauged space. The information transfer from gauged to ungauged basins, which is relevant to monitoring network design, represents a gap between hydrographic and LSH datasets which the BCUB dataset aims to fill.

## 1.2    Related datasets

### 1.2.1    Hydrographic Datasets

The BC Freshwater Atlas (FWA) (Gray, 2010) is the definitive source of freshwater feature mapping for British Columbia (BC). It contains roughly 3 million geometries representing the province-wide set of $1^{st}$ order fundamental component watershed units, with a reference system designed to facilitate aggregation into larger watershed assessment units. The FWA dataset is strictly limited to the administrative bounds of BC, cutting off many important trans-boundary basins at borders. Since the dataset is primarily hydrographic, it does not include catchment attribute information commonly used in rainfall-runoff model calibration. The FWA is provided with an open-use license, but the code used to derive the dataset is to our knowledge unpublished, and as such it isn't readily replicable or extensible with consistent input data and methodology.

The National Hydrographic Network (NHN) (Geobase, 2004) contains a hydrographic feature set similar to the BC FWA. It covers all of Canada and includes trans-boundary basins along the US border, but the geometries are organized in Work Unit Limits (WULs) which break up complete basins. The watershed attributes are similarly limited, and the code used to derive the geometries is to our knowledge unpublished.

HydroSHEDS is a dataset for global-scale applications featuring river networks, watershed boundaries and other hydrological features derived from the NASA Shuttle Radar Topography Mission (SRTM) DEM for most of North America at a resolution of roughly 90m. At latitudes $> 60°$ North, corresponding to the northern border of BC with the Yukon territory, HydroSHEDS catchments are derived from more coarse ($\approx 500m$) Hydro1k (Wickel et al., 2007) elevation data. Attributes derived from distinct elevation data sources are difficult to compare as discussed in subsection 2.2, as the stream networks (and catchment boundaries) are unique to a DEM source and to the data processing methodology (Datta et al., 2022). Studies using the HydroSHEDS dataset typically exclude catchments smaller than 100 km$^2$ (Guth, 2011; Zhang et al., 2020; Kratzert et al., 2023).

### 1.2.2    Large Sample Hydrology Datasets

Do et al. (2018) presents a review of key precedents in the emergence of LSH datasets, beginning with the Global Runoff Database (GRDB) which gathered daily and monthly streamflow time series observations from over 9000 catchments around the world. The MOPEX (Model Parameter Estimation Experiment) dataset, (Duan et al., 2006) which provides hydrometeorological time series data for 438 U.S. catchments, and the GAGES-II dataset (Falcone, 2011), which features geospatial attributes of monitored catchments, both laid the groundwork for the CAMELS dataset (Addor et al., 2017). CAMELS repre-

sents a combination of both attributes and hydrometeorological time series for a larger number of catchments in the U.S. The CAMELS approach has since expanded to other regions, with CAMELS-CL (Chile)(Alvarez-Garreton et al., 2018), CAMELS-BR (Brazil) (Chagas et al., 2020), CAMELS-AUS (Australia) (Fowler et al., 2021), and CAMELS-GB (Great Britain) (Coxon et al., 2020) published between 2018 and 2021. The HYSETS dataset (Addor et al., 2020) expanded continental scale LSH efforts across Canada, the US, and Mexico, combining catchment attributes with streamflow records and gridded climate products, contributing to the convergence of large-sample hydrology research. The recent Caravan dataset (Kratzert et al., 2023) standardizes and integrates these regional and global datasets into a unified resource, supporting large-scale hydrological modeling across diverse geographic contexts.

Large sample hydrology (LSH) datasets have excluded small basins, primarily due to uncertainties in basin delineation (Arsenault et al., 2020; Addor et al., 2017) and to ensure sufficient sample size for estimating attributes from gridded data sources at different resolution (Guth, 2011). However, the rationale for a specific threshold is generally not given. The HYSETS dataset flags basins smaller than $50 \text{ km}^2$, representing nearly one third of the dataset.

Regional datasets such as HYSETS and CAMELS often rely on catchment polygons sourced from official governing bodies, but the methods used for delineation are to our knowledge unpublished and likely vary in both approach and underlying data sources, leading to uncertainties in basin delineation and attribute estimation. This uncertainty highlights a gap that can be addressed in part with continuous and complete DEM coverage at higher resolution.

A large and diverse set of ungauged locations and associated attributes is sought to represent the decision space for monitoring network analysis and optimization, and more generally to support water resources research where catchment-based geospatial attributes are relevant.

## 1.3 British Columbia Ungauged Basin (BCUB) Database

The BCUB database contains a wide array of attributes describing the terrain, land cover, soil permeability and porosity, and climate of over 1.2 million (sub-)basins. We use the term 'basin' to refer to the local watershed of any confluence or outlet in a stream network, including individual upstream branches and their combination. Figure 1 shows the pour points representing the BCUB dataset, and the streamflow monitoring stations from the HYSETS dataset (Arsenault et al., 2020) that lie within the study region. The study region represents any terrestrial area within or upstream of any point within the BC administrative boundary (red dashed line in Figure 1), plus a buffer to include trans-boundary catchments and to mitigate the edge selection bias of optimal sensor placement in random fields (Hershfield, 1965; Rouhani, 1985; Krause et al., 2006).

The attribute set describing each sub-basin follows the HYSETS dataset as much as possible and includes select additional climate indices following the Camels dataset (Addor et al., 2017) to demonstrate how derived parameters can be added to the dataset. Three sets of land cover indices from the North American Land Change Monitoring System (NALCMS) (Latifovic et al., 2010) representing 2010, 2015, and 2020 are included to support questions about land cover change at the basin level as called for by Addor et al. (2020). An example plot showing forest cover change between 2010 and 2020 is shown in section 3.

Following Wilkinson et al. (2016), to support knowledge discovery, innovation, and integration of data and methods in subsequent work, both the data and the code used to generate the data are openly available. The code is provided not to

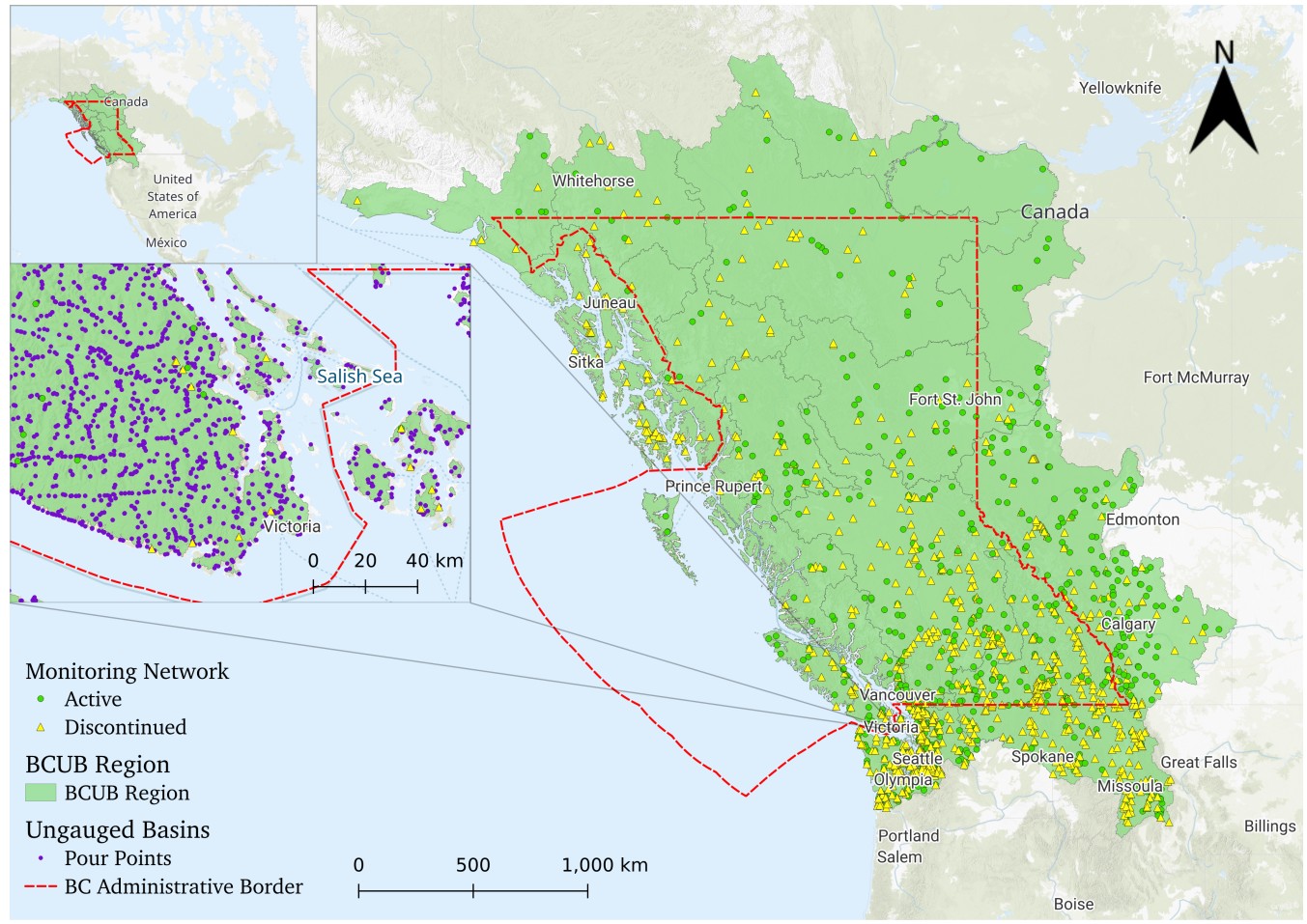

**Figure 1.** The study region (right) expands beyond the British Columbia administrative border to capture trans-boundary regions. Active and discontinued streamflow monitoring stations (those included in HYSETS (Arsenault et al., 2020)) are sparse and unevenly distributed as shown in the main figure at right, and the inset detail shows the high density of pour points (purple) defining catchments in the BCUB dataset. (basemap from © MapTiler © OpenStreetMap contributors)

champion a particular method, but to highlight the nuance involved in developing large sample datasets that for brevity and clarity are generally left out of dataset description papers. There are no stochastic elements in the methodology, yet there are a large number of methodological choices that yield distinct outcomes. Providing the complete code at minimum aims to be explicit about these choices.

Our goal with the BCUB dataset was to provide a representative set of catchment attributes that cover key groups commonly found in the literature–terrain, land cover, climate, and soil. While the attribute set is not as extensive as those found in the LSH literature, we prioritized creating a transparent, extensible data product with complete code and tutorial-like supporting

**Table 1.** Summary of catchment attribute source data.

| Dataset | Attributes | Source |
|---|---|---|
| USGS 3DEP[1] | **Terrain**: area, elevation, aspect, slope | (U.S. Geological Survey, 2022) |
| GLHYMPS[3] | **Soil**: porosity, permeability | (Huscroft et al., 2018) |
| NALCMS[2] | **Land cover (2010, 2015, 2020)**: forest, shrubs, grassland, wetland, crops, urban, water, snow and ice | (Latifovic et al., 2010) |
| DAYMET[4] | **Climate (daily estimates, 1980-2022)**: precipitation, temperature, snow water equivalent, vapour pressure, shortwave radiation, and duration and frequency of high and low precipitation | (Thornton et al., 2022) |

1. 3DEP: 3D Elevation Program, U.S. Geological Survey,

2. NALCMS: North American Land Change Monitoring System, accessed at http://www.cec.org/north-american-land-change-monitoring-system/

3. Global Hydrogeology Maps.

4. Gridded daily climate estimates on a 1-km Grid for North America, Version 4. https://daymet.ornl.gov/

information. Given the rapid development of attributes in LSH research, we focused on providing a solid framework rather than the most exhaustive or up-to-date set of attributes.

## 2 Data & Methods

### 2.1 Data collection and pre-processing overview

Attributes of ungauged basins were clipped from the digital elevation, land cover, soil, and climate geospatial datasets described in Table 1 through a data preparation and processing pipeline described in Figure 2. Individual catchment polygons were delineated from the set of pour points in the stream

network representing river confluences. The stream network was derived from the 1 arc-second (30m at the equator) resolution USGS 3DEP (U.S. Geological Survey, 2022) digital elevation model (DEM) using the open-source software library Whitebox (version 2.3) (Lindsay, 2016). Streams are

defined by a minimum upstream accumulation of 1 km$^2$ to match the smallest monitored catchment in the HYSETS dataset.

The study region was divided into complete basin subregions (no surface inflow across boundaries) as shown in

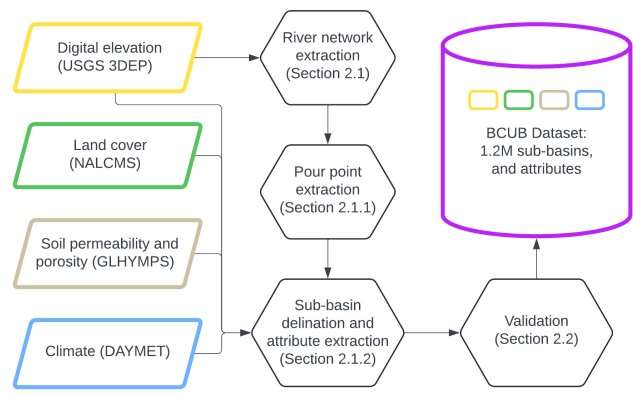

**Figure 2.** Schematic of the BCUB development pipeline, from retrieving input datasets from external sources to creating a final database of sub-basins and their representative catchment attributes.

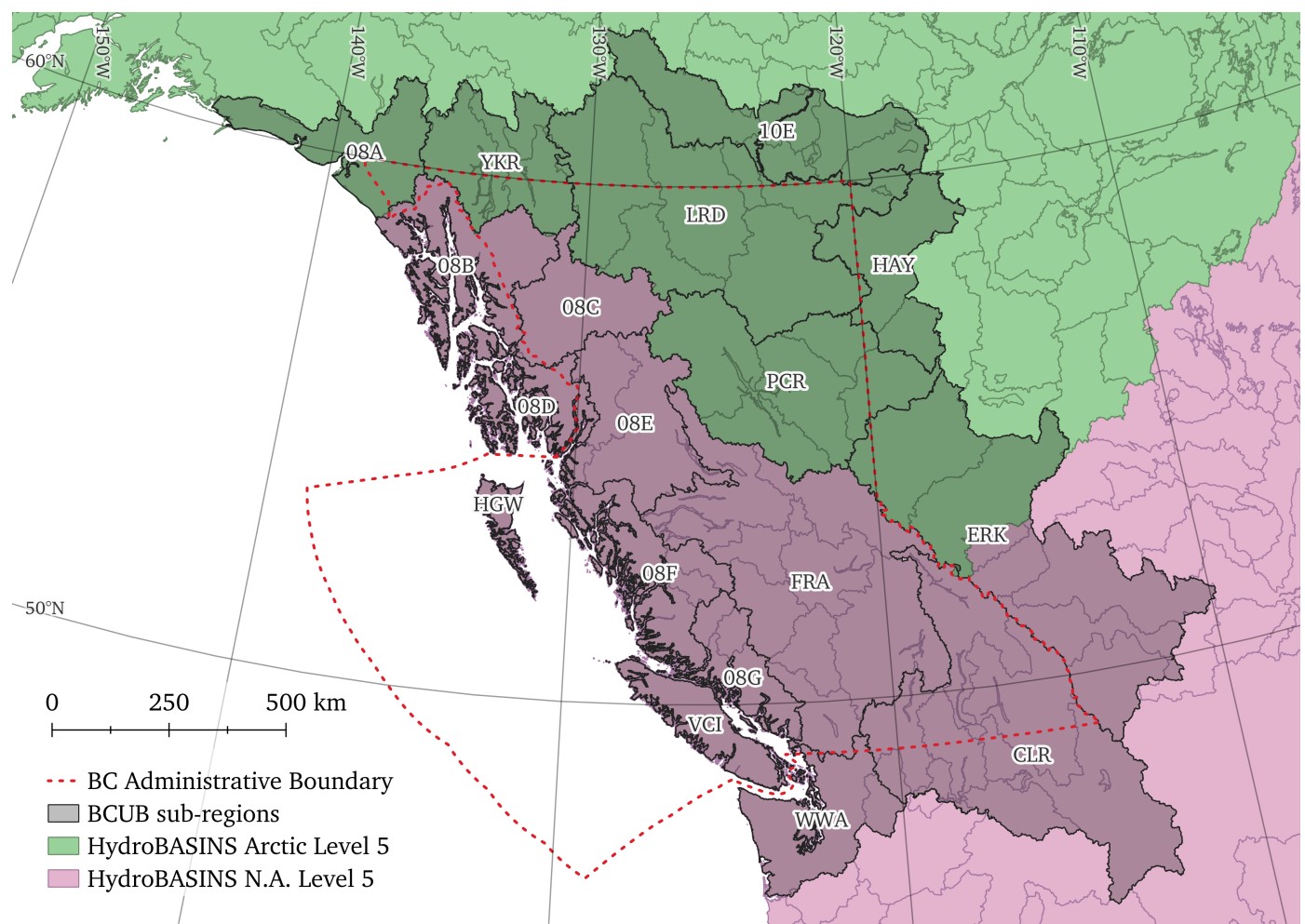

**Figure 3.** At right, the study region is divided into complete watershed sub-regions (encoded in the "region_code" parameter) by merging level 5 & 6 HydroBASINS polygons to cover the BC boundary. The study region extends beyond the administrative border of BC to include trans-boundary basins and a minimum buffer of $\approx 100$ km. The purpose of merging complete watershed regions is to manage computational resources the DEM pre-processing $\rightarrow$ sub-basin delineation $\rightarrow$ attribute extraction pipeline.

Figure 3 (right) assembled from HydroBASINS (Lehner et al., 2021) data to simplify the automated sub-basin delineation and attribute extraction work flow. The data processing pipeline is described as follows:

1. **Define study region and sub-regions**: Level 5 and 6 watersheds from the HydroBASINS dataset were used as a first approximation to break the study region into smaller components for memory management in data pre-processing. Study region bounds were refined by deriving the covering set of basins in each region independently, see subsubsection 2.2.1
for more detail about the treatment of region bounds.

2. **Retrieve DEM data**: The study region bounding box was used to download the covering set of digital elevation tiles from the USGS 3D Elevation Program (U.S. Geological Survey, 2022). In addition, lower resolution (90m) DEM tiles from EarthEnv DEM90 (Robinson et al., 2014) were used in the data validation analysis presented in subsection 2.2.

3. **Pre-process DEM raster**: Hydraulic conditioning of the DEM, including depression filling, resolving flats, computing flow direction and accumulation, and stream network extraction were processed using the open-source geospatial analysis software Whitebox (Lindsay, 2016).

4. **Define and filter pour points**: Pour points define the outlet of each catchment and their precise location is specific to the input DEM and pre-processing steps. Each ungauged catchment is delineated from a pour point defined by the stream network. Lake polygons from HydroBASINS were used to filter out pour points within lakes. Points are flagged (*in_perennial_ice*) where the 2020 NALCMS land cover classification is perennial ice and snow.

5. **Catchment delineation**: Catchment polygons were derived from sets of input pour point coordinates using the "Unnest-Basins" function in Whitebox.

6. **Attribute extraction**: Catchment polygons were used as clipping masks to capture representative values from the various geospatial layers. Attribute indices were aggregated from raster and vector layers as described in Table 2.

Additional detail about pour point selection, catchment attribute extraction, and data processing follows.

### 2.1.1 Pour point set selection

The sub-basins in the BCUB database are delineated from a subset of raster cells representing the stream network. The set of pour points points used for catchment delineation is called the *candidate monitoring location* (CML) set. By limiting the CML set to river confluences, the number of polygons to process is reduced to < 5% of the complete set of stream network cells. Since changes in upstream accumulated area are small along reaches between confluences, and by extension changes in the hydrologic properties of the sub-basin are small, eliminating these points reduces redundancy and data processing.

The CML set is defined by the following criteria:

1. **Confluences**: stream cells with more than two neighbouring stream cells (8-direction grid), where the flow direction of more than one neighbouring stream cell is pointed toward the target cell, and

2. **River outlets**: intersections of river network lines with ocean coastline, major regional watershed outlets at the study region boundary, and confluences with lakes where the upstream contributing area is at least $1 \text{ km}^2$.

Stream confluences within lakes were excluded from the pour point set, as illustrated in Figure 4 where a red "x" denotes a confluence in the spurious network within a lake, a yellow triangle represents the location where a river drains into a lake. Green circles represent confluences and individual upstream branches.

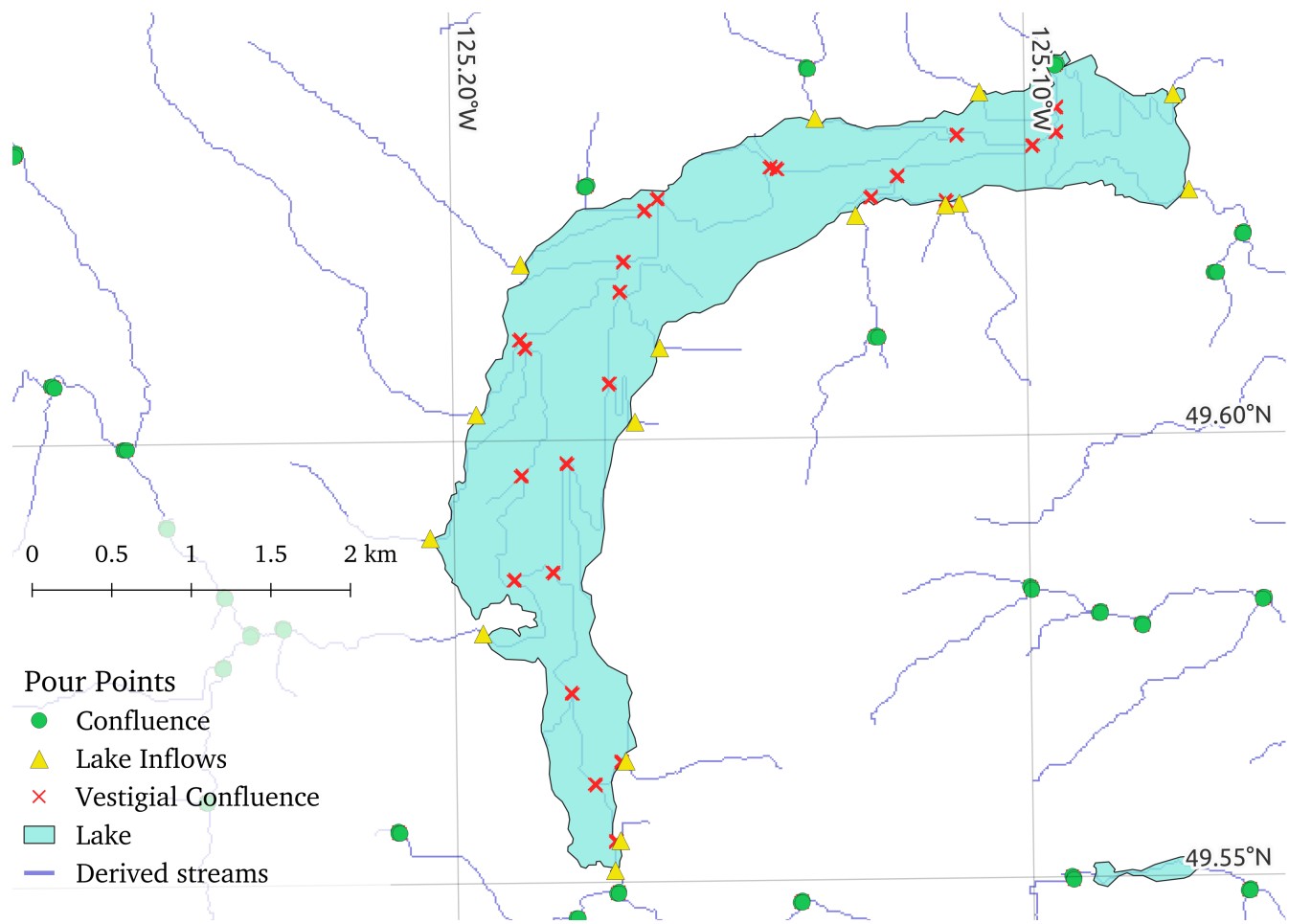

**Figure 4.** In the pour point identification algorithm, the stream network (blue lines) derivation creates spurious confluences within lakes (red "x") which are excluded, and river-lake confluences (yellow triangle) are added. Stream confluence points (green circles) are the upstream branches of converging streams and their combination.

The headwaters mapped in the stream network are simply a vestige of the minimum area threshold ($1~\text{km}^2$) used to define a stream network, so they are excluded from the pour point set. Accurate headwater identification (network extent mapping) requires a more rigorous approach to address uncertainty related to stream permanence (Shavers and Stanislawski, 2020). Mutzner et al. (2016) found classical (i.e. cumulative drainage area) threshold approaches do not capture spatial variability of headwater drainage networks in mountainous regions compared to detailed field survey mapping, and statistical methods

are likewise unable to resolve local topography to accurately map headwater streams at low-resolution. Further discussion of uncertainty is provided in subsection 2.2.

### 2.1.2 Sub-basin delineation and notes on attributes

A catchment boundary polygon was generated for each pour point in the CML set using the "unnest basins" function in the Whitebox software library (Lindsay, 2016). Attributes were derived for each sub-basin by i) using the polygons as raster clipping masks, and ii) spatial intersection of the polygon and geospatial raster and vector data in PostGIS (PostGIS Project Steering Committee and others, 2018).

Attribute values were computed using the geometric mean of the raster pixel values contained in basin polygons in the case of soil permeability, the circular mean in the case of slope aspect, the fraction of total area in the case of land use, and the spatial mean for all other attributes. Physical attributes are described in Table 2, and metadata attributes are described in Table 3.

Several binary attributes are included in the attribute set to represent uncertainty in geometry and value estimates. A 'soil_flag' value of 1 indicates that the clipped soil data differs from the catchment polygon area by more than 5% to indicate gaps in the GLHYMPS (soil) data. A 'permafrost_flag' value of 1 represents the presence of permafrost in the basin. A value of 1 for the 'in_perennial_ice' flag represents a pour point location where the land cover classification is "perennial snow and ice" as defined by (Latifovic et al., 2010). A 'geometry_flag' value of 1 represents a catchment intersecting or touching an uncertain area along the region boundary whose area is >= 5% of the catchment area, as described in subsubsection 2.2.1.

### 2.1.3 Data processing notes

Beyond data sources, the offline approach of deriving sub-basins from source data and writing code to process attributes was adopted despite the elegant online polygon aggregation and processing approach demonstrated by Kratzert et al. (2023) in developing the Caravan dataset with use of Google Earth Engine (GEE) (Gorelick et al., 2017). Such an approach is preferable from the perspective of standardized methods of catchment attribute extraction, but for our target of ungauged catchments it does not eliminate the need for DEM pre-processing to generate stream networks, for filtering and extracting pour points, or for sub-basin delineation. These steps represent a substantial portion of the attribute extraction workflow, and what remains to process with GEE is still subject to usage limits, namely for processing the very large set of polygons, even considering an aggregated polygon approach.

A benefit of the offline approach is generating a set of sub-basin polygons from the highest resolution DEM available that is continuous and complete, and ensuring that basin polygons match the DEM source from which terrain attributes are derived.

Expansion of the study region or addition of new attributes can be accomplished by following the processing methodology in the code repository provided. Four parameters derived from the Daymet daily precipitation data are processed in the code provided do demonstrate how computed parameters can be added to the BCUB from existing input data. The examples follow the Camels dataset Addor et al. (2017) and include:

1. **Low precipitation frequency**: frequency of days where precipitation $< 1 \, \mathrm{mm \, day^{-1}}$),

2. **Low precipitation duration**: average duration of low precipitation events, or the number of consecutive low precipitation days $< 1 \, \mathrm{mm \, day^{-1}}$,

3. **High precipitation frequency**: frequency of days where precipitation is $\geq 5$ times the mean daily precipitation, and

4. **High precipitation duration**: average duration of consecutive high duration events, number of consecutive high precipitation days $\geq 5$ times mean daily precipitation.

## 2.2 Technical Validation

The large number of geometries in the BCUB dataset requires an automated approach to validate the sub-basin polygons used to capture attributes. The representativeness of attributes is a function of the accuracy of the stream network derived from DEM. Higher resolution DEM can better resolve lower-relief topographic features resulting in better basin delineation performance, particularly for small basins (Zhang and Montgomery, 1994; Tarolli and Dalla Fontana, 2009; Woodrow et al., 2016).

It is important to emphasize that the $1 \text{ km}^2$ minimum drainage area threshold introduces significant uncertainty in the accuracy of the smallest sub-basins, and those where topographic relief is low. Detailed validation of stream network accuracy is left to future work that the BCUB is intended to support, and validation of the smallest sub-basins used in studies is left to the user. Next we discuss indirect attribute validation methods, and limitations of the dataset and methods.

### 2.2.1 Region boundary treatment

While the region polygons assembled from HydroBASINS are a helpful tool for organizing the data processing pipeline, the resulting bounds are different from those produced by independently delineating basins from the 1 arc-second DEM used in this study. These differences are comparable in size to the smallest sub-basins in the BCUB dataset, introducing uncertainty into the attributes of any catchment whose boundary touches or intersects them. Boundary deviations are defined as i) gaps between region bounds where the DEM does not resolve an outlet, and ii) boundary overlaps between regions with shared boundaries.

The Caravan dataset (Kratzert et al., 2023) clearly describes the issue with aggregating attributes from catchment boundary polygons that do not precisely align with the HydroBASINS polygons. By independently deriving the region bounds from a single continuous DEM source (1 arc-second USGS 3DEP), we avoid the problem of misalignment with HydroBASINS polygon. This process does not guarantee perfect alignment of region bounds, but the mean size of deviations is significantly reduced.

The edge detail inset Figure 2.2.1 shows an example segment of region boundaries aggregated from HydroBASINS (blue dashed line) compared to those derived from the USGS 3DEP (1 arc-second) DEM. In Figure 2.2.1, the purple (Peace, PCR) and green (Fraser, FRA) areas represent the (BCUB region) boundaries delineated from the 1 arc-second DEM. White areas are gaps that remain following the iterative boundary definition process described below.

To avoid restricting the catchment boundary delineation by the clipping mask, a (5 km) buffer was applied to the region boundaries aggregated from level 5 and 6 HydroBASINS polygons. The buffered polygons were used as clipping masks on the DEM before deriving the covering set of polygons (catchments) for each region. The covering set is defined as the smallest number of non-overlapping polygons covering a region. The exterior edges (of the union of intersecting geometries) were checked to verify that they do not touch the edge of each buffered region polygon. Where the edges intersect, the buffer (DEM

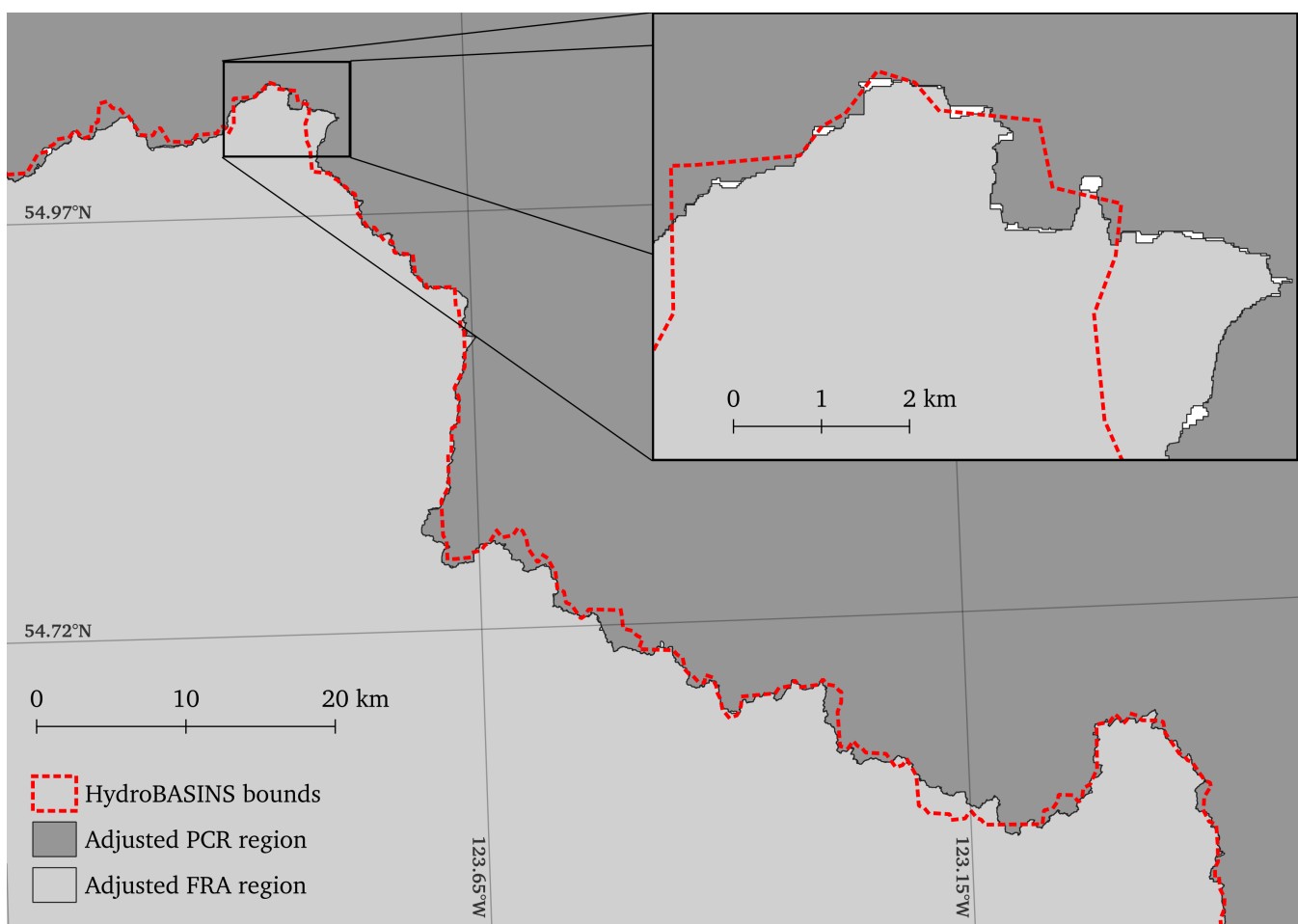

**Figure 5.** An example edge detail showing the disagreement between region bounds derived from different sources. The red dashed line represents the HydroBASINS bounds while the shaded areas represent sub-regions delineated independently from USGS 3DEP DEM. The higher resolution DEM resolves more topographic detail yielding different bounds, but neighbouring region bounds do not exactly match, as can be seen by the gaps remaining in the inset detail.

clipping mask) was manually expanded in QGIS and the process repeated until the buffer was sufficient, i.e. the covering set of basins does not touch the edge of the clipping mask. The use of a buffer produces small peripheral catchments draining to adjacent region basins, and these are excluded by identifying that they are completely contained by the clipping mask of the adjacent regions.

Delineating region boundaries independently from the HydroBASINS polygons does not yield perfectly shared boundaries, but the resulting deviations are substantially smaller. The distribution of the size of deviations from shared sub-region boundaries is shown in Figure 6. The red series represents differences between the BCUB region bounds and HydroBASINS-derived bounds (median area of $0.13\text{km}^2$), while the blue series represents disagreement (overlaps and gaps) between the BCUB sub-

265 region boundaries (median area 0.025km$^2$). Polygons smaller than 0.01 km2, or 1% of the smallest sub-basin in the BCUB dataset were neglected. The boundary deviation polygons (gaps and overlaps) are included in the code repository.

The uncertainty introduced by missing or overlapping areas along sub-region bounds is addressed in the BCUB dataset in two ways. The 'geometry_flag' attribute indi-
270 cates that a catchment polygon intersects or touches an uncertain region bound if the total uncertain area represents at least 5% of the catchment area. Where catchments derived from distinct basin outlets overlap, either catchment may overestimate the area, and where an area is not cov-
275 ered by any basin but is not necessarily endorheic, either bordering sub-basin may underestimate the catchment area. Where a catchment polygon touches or intersects with an uncertain boundary, the size of the uncertain area is represented by a positive integer value to indicate potential over-
280 estimation ('inside_pct_area_flag') or underestimation ('outside_pct_area_flag') of the catchment as a percentage of the catchment area. The purpose of including these quantities is to identify and express the significance of uncertain catchment bounds.

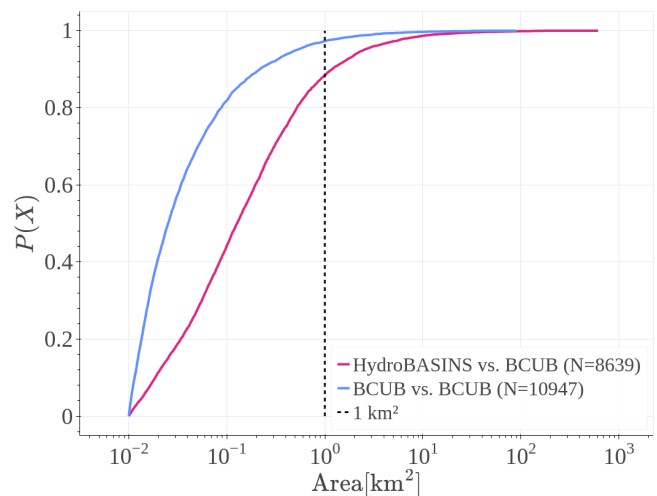

**Figure 6.** Distributions of geometric deviations (uncertain edges) between shared sub-region boundaries from HydroBASINS-based region polygons (red series, median area 0.13 km$^2$) and the improvement from deriving region bounds (blue, median area 0.025 km$^2$).

### 2.2.2 Vestigial effects of DEM resolution

In addition to the hydraulic conditioning process for stream network derivation, the grid representation of elevation introduces vestigial artifacts in the representation of basins, and consequently, catchment attribute estimates.

The stream network derived from DEM does not capture permanent water bodies, resulting in spurious river confluences. These vestigial confluences were excluded by using the lakes geometry layer from HydroBASINS as a mask, as described in
subsubsection 2.1.1. Since HydroBASINS is derived from different sources, hydrographic features do no align exactly with the stream network we derived from the 1 arc-second DEM.

The disk space required to store a polygon is a linear function of the number of vertices defining it, and the precision of geographic coordinates describing the geometry. The sub-basin polygons are simplified (using the Shapely library (Gillies, 2021) "simplify" function) using a tolerance equal to one-half the diagonal length of the raster pixel resolution. Simplifying
(or smoothing) polygons represents a trade-off between reducing the disk and bandwidth required to store and transmit large sets of geometries, and the representativeness of attributes that are captured by intersecting each polygon with the various geospatial raster layers. The effect of polygon simplification is discussed in more detail in subsubsection 2.2.3.

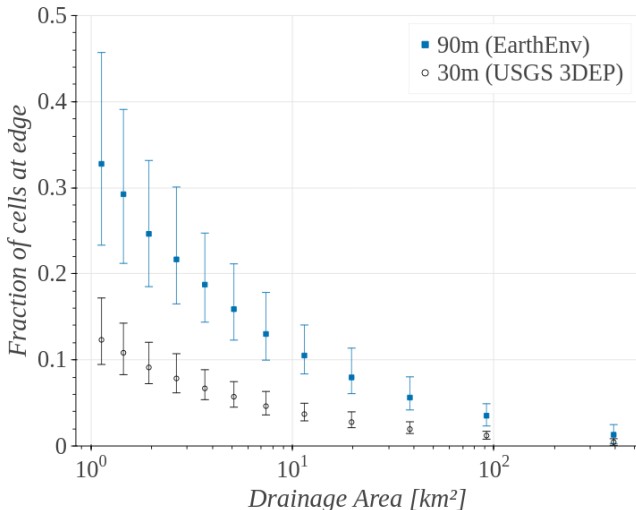

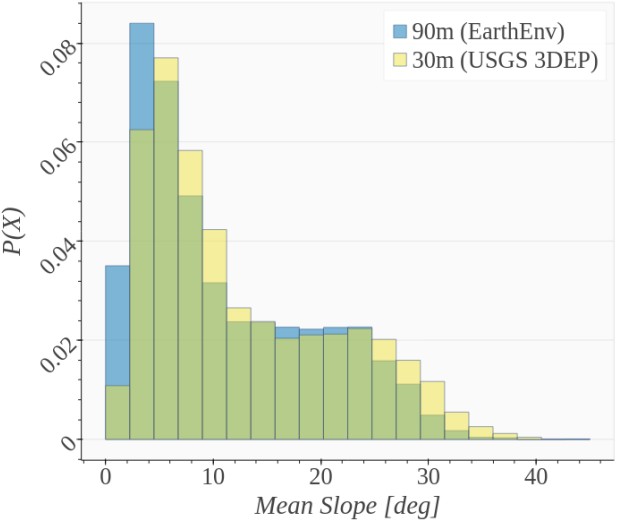

**Figure 7.** As the drainage area decreases, the number edge pixels becomes a significant proportion of the total number of pixels representing the sub-basin. Points in the above figure represent bin median values based on equiprobable binning ($N \approx 600$ samples per bin), and the whiskers represent the 90% confidence interval.

**Figure 8.** Higher resolution DEM captures greater topographic relief as shown by comparing the distribution of mean slope between 30m (USGS 3DEP) and 90m (EarthEnv) DEM on a random sample of 10,000 basins.

The set of raster pixels representing each sub-basin is captured using the "crop-to-cutline" function from the open source GDAL library (GDAL/OGR contributors, 2023) which by default captures pixels whose centroid lies within the polygon (pixels are not points, but quadrilaterals). Alternatively the larger set of *intersecting* pixels can be selected by setting the "CUTLINE_ALL_TOUCHED=TRUE" keyword argument. As drainage area decreases (or raster resolution decreases), the difference in edge pixel selection method represents an increasing proportion of total pixels which may then yield significant differences in attribute values depending upon the clipping method used. Figure 7 shows that the proportion of edge pixels representing the catchment increases with decreasing area, and uses the USGS 3DEP (30m grid at the equator) and EarthENV DEM90 (EENV) DEM (90m grid at the equator) to show how the proportion of edge pixels changes with DEM resolution. The purpose of this exercise is to highlight one source of uncertainty introduced by the data processing methodology and to demonstrate the effect of the clipping method as a function of catchment scale.

Mean slope is a widely used attribute in large sample hydrology (Addor et al., 2017; Alvarez-Garreton et al., 2018) to describe the degree of topographic relief of a catchment, defined in Arsenault et al. (2020) as "the average slope when considering the individual elevation differences between tiles" (raster pixels). We used WhiteboxTools to compute the slope of each DEM pixel using a 3rd-order Taylor polynomial fit (Florinsky, 2016) with a kernel size of 5x5 pixels. Mean catchment slope increases with increasing resolution because topographic relief is better captured at higher resolution (Zhang and Montgomery, 1994). Figure 8 compares mean slope between 30m and 90m resolution DEM sources, where the higher resolution DEM is able to resolve greater topographic detail. The comparison is based on a random sample of roughly ten thousand polygons

in the BCUB dataset ranging in size from 1 km$^2$ to $2 \times 10^5$ km$^2$. The sample of sub-basins in Figure 8 shows a bias toward lower calculated mean slope from the lower resolution DEM source using the same polygon mask to capture pixels. Further interpretation of these differences is left to future work.

### 2.2.3 Catchment Attributes and Self-Similarity

Mandelbrot (1967) described the measurement of coastline length as a function of the scale of observation, and the lines describing features like catchment boundaries and stream networks also exhibit self-similarity. Perimeter, stream gradient, and shape factors like elongation or compactness are length-based attributes used in many LSH datasets (Arsenault et al., 2020; Klingler et al., 2021; Kratzert et al., 2023). The compactness coefficient is defined as the ratio of polygon perimeter to the circumference of a circle with equal area ((Gravelius, 1914) as cited in (Sassolas-Serrayet et al., 2018)) . Length-based attributes are not comparable without consistent input DEM resolution and data pre-processing.

The difference in catchment boundary lines shown in Figure 2.2.3 illustrates why perimeter measurement can vary considerably due to input DEM resolution or catchment delineation methodology. Perimeter is not included in the BCUB attribute set because unless otherwise treated, polygons derived from higher resolution DEM will measure a longer perimeter.

In July 2022 the Water Survey of Canada published updated catchment boundaries representing the majority of the streamflow monitoring network. These updated geometries can be accessed at the (WSC) National Water Data Archive. We found all polygons common to both the HYSETS dataset and this updated polygon set, and computed pairwise comparisons of perimeter lengths. There were 1035 sub-basin polygon revisions that did not meet the similarity criteria, reflecting the difficulty in retrospectively defining streamflow monitoring station locations from historical records (Arsenault et al., 2020).

The sample used for the perimeter comparison includes 715 sub-basins where the original and updated polygons were a close match to control for significant changes in the

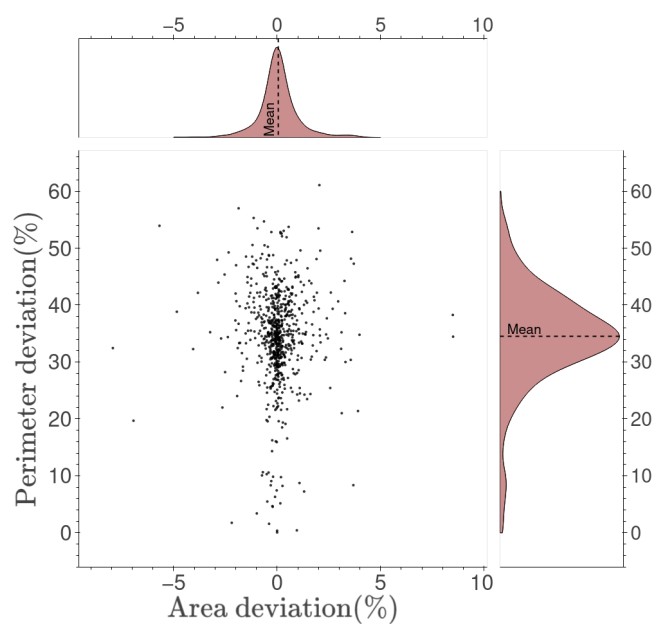

**Figure 9.** The DEM resolution and catchment delineation method affects the measurement of perimeter. Comparing revisions of catchment boundaries representing the same streamflow monitoring stations, the perimeter length is significantly different despite the area being nearly constant, and despite a close match between polygons according to a Jaccard Similarity Index match of $\geq 95\%$.

polygon shape. A "close match" is defined as the ratio of intersecting area to union area (Jaccard similarity index) $\geq 95\%$. Figure 9 shows the newer revision polygon perimeter measurements are substantially greater, and the deviation exists indepen-

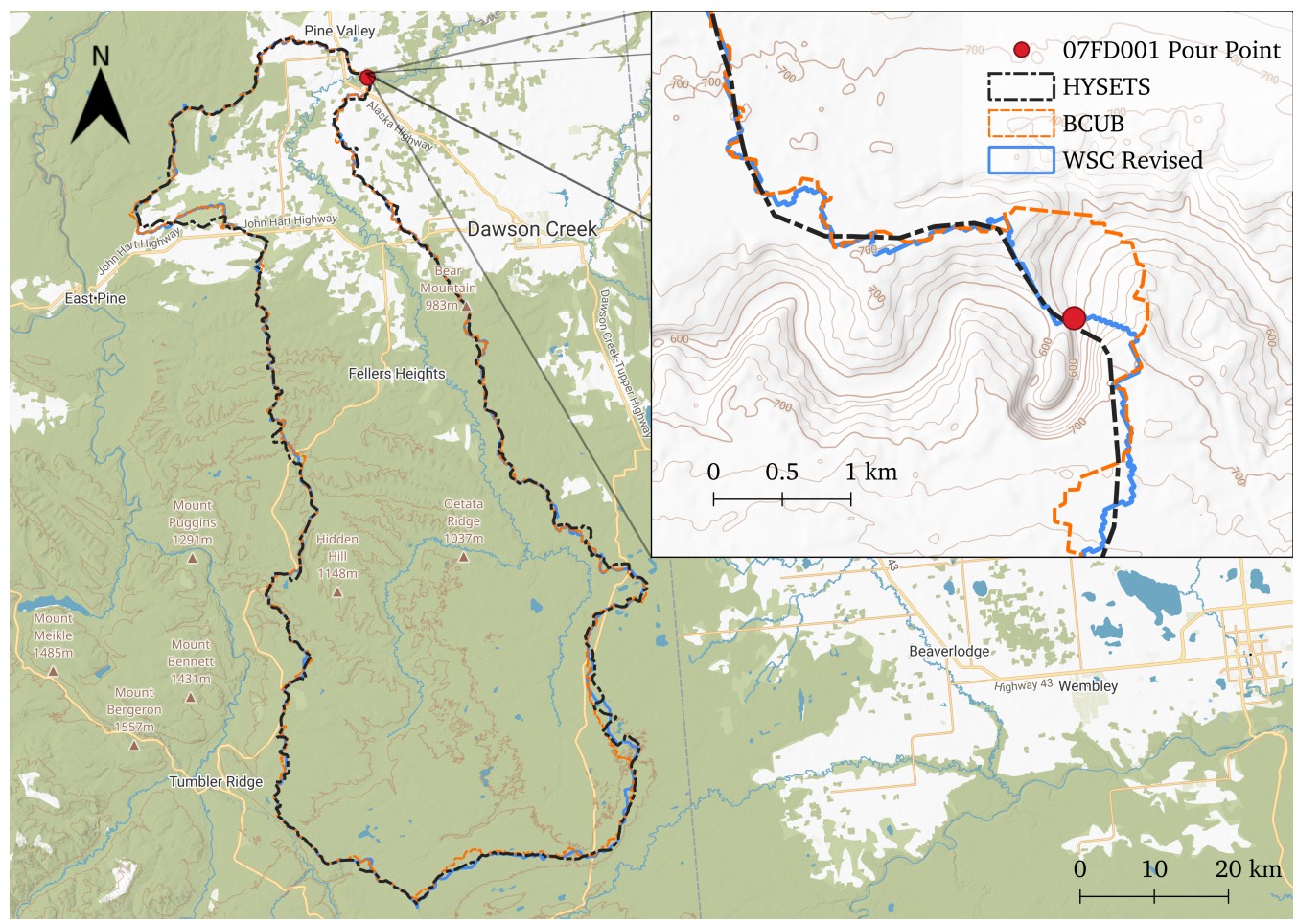

**Figure 10.** An example edge detail of the same catchment boundary from three different sources where the intersecting area is over 98% of the published value. The HYSETS dataset polygon (back dash-dot line) comes from an earlier revision published by the WSC representing the Kiskatinaw River near Farmington (WSC ID 07FD001), while a recent revision (July 2022) by the WSC (solid blue) shows a distinct difference in polygon edges. The polygon from the BCUB (dashed orange) derived from USGS 3DEP DEM is different from both. (basemap from © MapTiler © OpenStreetMap contributors)

dent of spatial scale. This difference highlights the need to ensure consistent input DEM and data processing methodology if length-based attributes are included attribute datasets.

Average stream gradient is a length-based attribute that is a function of both raster resolution and the assumed location of channel head, usually by minimum area threshold. Robinson et al. (2014) calculated mean stream gradient as the ratio of the maximum total elevation change in the basin stream network to the length of the corresponding river reach. Stream length is a function of DEM resolution, and the length of reach is measured from the catchment outlet to an uncertain headwater location (Hafen et al., 2020, 2022). In the derivation of the stream network for the BCUB dataset, headwater locations are simply a

vestige of the assumed minimum drainage area threshold, and as a result an attribute representing average stream gradient is not included in the BCUB database.

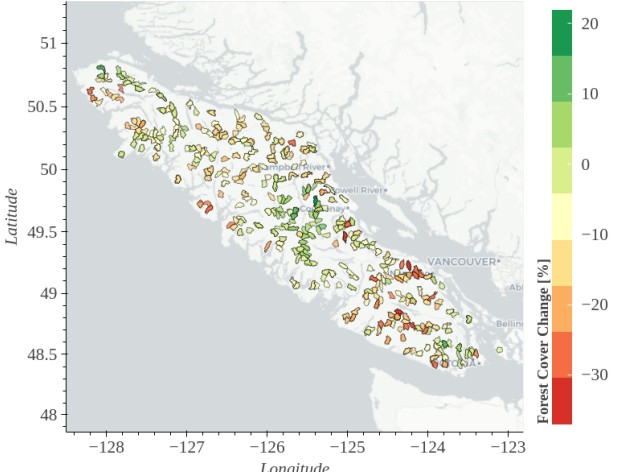 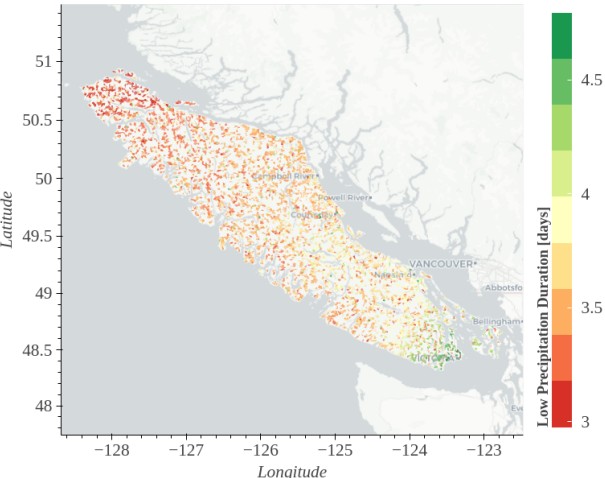

**Figure 11.** Example visualization using the BCUB dataset maps the % change in forest cover (as a percentage of the catchment area) for sub-basins with drainage area between 20 and 25 km² on Vancouver Island (VCI). Basemap from © OpenStreetMap contributors. Plotted using the Bokeh visualization library in Jupyter notebooks.

**Figure 12.** Example visualization using the BCUB dataset maps the mean annual duration of low precipitation (< 0.1mm/day) for catchments with drainage area between 2 and 5 km² on Vancouver Island (VCI). Basemap from © OpenStreetMap contributors. Plotted using the Bokeh visualization library in Jupyter notebooks.

## 3 Usage Notes

It is the hope that the BCUB dataset will serve a wide range of water resource research and practice where catchment-based attributes are integral to the methodology, or perhaps more importantly to express the limits of appropriate use and interpretation. Figure 11 and Figure 12 provide two basic examples of the kind of sub-basin level querying the BCUB is designed to support. Figure 11 shows catchment-level changes in forest cover between 2010 and 2020 for basins in the range of 20 to 25 km², and Figure 12 shows the mean duration of dry periods (days with less than 0.1 mm rainfall) for catchments between 2 and 5 km².

Stream networks are unique to the input DEM, and they are affected by the choice of pre-processing steps. The greatest degree of uncertainty is associated with the smallest catchments with the lowest topographic relief. Zhang and Montgomery (1994) provides guidance about interpreting features at scales relative to DEM resolution. The representativeness of stream networks, and by extension the attributes captured by polygon masks generated from stream networks, is an important component of uncertainty analysis and data reliability assessment. This aspect of the analysis is left to future work that the BCUB dataset is designed to support, in particular the lower limit of basin scale that can be supported by 1 arc-second DEM.

## 4 Code and data availability

The BCUB dataset (Kovacek and Weijs, 2023) is accessible under a Creative Commons BY 4.0 license through the Borealis data repository at https://doi.org/10.5683/SP3/JNKZVT. A summary of the dataset contents and supporting information is presented in Table 4. The sub-basin polygon geometries are provided in the open-source, cross-language Apache Parquet format (https://parquet.apache.org/), which has the convenience of supporting multiple geometries. The Parquet file format is supported by several widely used Python libraries, including Dask (https://docs.dask.org/) and GeoPandas (https://geopandas.org/), and the Arrow package features an interface for the R programming language (https://arrow.apache.org/docs/r/). The dask-geopandas library in Python (https://dask-geopandas.readthedocs.io/) is recommended for performance with large datasets.

The catchment attributes are provided in two forms in the Borealis data repository. The larger form includes catchment boundary, centroid, and pour point geometries. These are saved in the Parquet file format under the 'basin_polygons' folder (select the "tree" view for easier navigation). The Parquet file naming convention follows the sub-region codes shown in Figure 3. A "light" format without geometries is provided in comma delimited format in BCUB_attributes_20240630.csv. Sub-region geometries with their associated codes are provided for reference in BCUB_regions_4326.geojson. Metadata describing the dataset is provided in MetaData.pdf, and additional sub-basin attribute information, including descriptions and sources is provided in the Readme.pdf.

The scripts used to derive the dataset, and the validation results and figures shown in this paper are provided in an open-source Github repository (https://github.com/dankovacek/bcub). The code to replicate the figures in subsection 2.2 is provided in the "validation" folder of the repository. Figures 1 to 3 and 6 were prepared with the QGIS software (QGIS Development Team, 2023), and all remaining figures were created using the Bokeh data visualization library (Bokeh Development Team, 2023) in Python.

In addition, an example guide is provided (https://dankovacek.github.io/bcub_demo/) through a set of Jupyter (Kluyver et al., 2016) notebooks to demonstrate the complete process of data retrieval, pre-processing, sub-basin delineation, attribute extraction, and data product usage. The code to produce Figure 11 using the Parquet file format is demonstrated in the final chapter of the Jupyter book demo, titled "Data Import and Usage Examples".

*Author contributions.* Daniel Kovacek wrote the code to create the dataset and the Jupyter Notebook tutorials, and wrote the manuscript. Steven Weijs provided research supervision and manuscript review.

*Competing interests.* The authors declare no competing interests.

*Acknowledgements.* This study received financial support from the British Columbia Ministry of Environment and Climate Change Strategy (Agreement #TP23EPEMA0031MY). The authors wish to thank the two anonymous reviewers whose feedback improved the quality of this manuscript. The authors also wish to express gratitude to all those contributing to open-source scientific software.

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

**Table 2.** Basin attributes in the BCUB database derived from USGS 3DEP (DEM), NALCMS (land cover), GLHYMPS (soil), and NASA Daymet (climate) datasets.

| Group | Description (BCUB label) | Aggregation | Units |
|---|---|---|---|
| | Drainage Area (drainage_area_km2) | at pour point | $km^2$ |
| Terrain | Elevation (elevation_m) | spatial mean | $m$ above sea level |
| | Terrain Slope (slope_deg) | spatial mean | ° (degrees) |
| | Terrain Aspect (aspect_deg) | circular mean[2] | ° (degrees) |
| | Cropland (land_use_crops_frac_<year>) | | |
| | Forest (land_use_forest_frac_<year>) | | |
| | Grassland (land_grass_forest_frac_<year>) | | |
| Land Cover[3] | Shrubs (land_use_shrubs_frac_<year>) | spatial mean | % cover |
| | Snow & Ice (land_use_snow_ice_frac_<year>) | | |
| | Urban (land_use_urban_frac_<year>) | | |
| | Water (land_use_water_frac_<year>) | | |
| | Wetland (land_use_wetland_frac_<year>) | | |
| | Permeability (logk_ice_x100) | geometric mean | $m^2$ |
| Soil[4] | Std. Dev. Permeability (k_stdev_x100) | geometric mean | $m^2$ |
| | Porosity (porosity_x100) | spatial mean | % cover |
| | Annual Precipitation (prcp) | | mm/year |
| | Daily Minimum Temperature (tmin) | | Celsius |
| | Daily Maximum Temperature (tmax) | | Celsius |
| Climate[5] | Annual Maximum Snow Water Equivalent (swe) | | Celsius |
| | Shortwave Radiation (srad) | spatial and | $W/m^2$ |
| | Vapour Pressure (vp) | temporal mean | Pa |
| | High precipitation frequency (high_prcp_freq) | | days/year |
| | Low precipitation frequency (low_prcp_freq) | | days/year |
| | High precipitation duration (high_prcp_duration) | | days |
| | Low precipitation duration (low_prcp_duration) | | days |

1. Spatial aspect is expressed in degrees counter-clockwise from the east direction.

2. The <year> suffix specifies the land cover dataset (2010, 2015, or 2020),.

3. Soil parameters follow definitions from Huscroft et al. (2018).

4. Only the climate parameters directly extracted from distinct Daymet source variables are shown here. Additional computed parameters are discussed in subsubsection 2.1.3.

5. A high precipitation event is defined as total daily precipitation greater than 5x the annual mean, and the duration refers to the mean duration of high precipitation events.

6. A low precipitation event is defined as total daily precipitation less than 0.1mm, and the duration refers to the mean duration of low precipitation events.

**Table 3.** BCUB dataset metadata attributes.

| Group | Description (BCUB label) | Units |
|---|---|---|
| | Region code identifier (region_code) | - |
| | Pour point[1] (ppt_x, ppt_y) | m |
| Metadata | Basin centroid[1] (centroid_x, centroid_y) | m |
| | Soil Flag (soil_flag) | binary (0/1) |
| | Permafrost Flag (permafrost_flag) | binary (0/1) |
| | Geometry Flag (geometry_flag) | binary (0/1) |
| | Geometry underestimation flag (outside_pct_area_flag) | % |
| | Geometry overestimation flag (inside_pct_area_flag) | % |

1. Geometries are projected to the BC Albers (EPSG:3005) coordinate reference system.

**Table 4.** Summary of data repository contents.

| Filename | Description |
|---|---|
| **BCUB_attributes_20240630.csv** | Catchment attributes with geographic coordinates describing the catchment centroid and the outlet (pour point). Catchment polygon geometries are not included for performance. |
| **polygons/*.parquet** | Basin attributes and associated catchment boundary, centroid, and pour ppoint geometries are organized into sub-regions to limit file sizes. |
| **BCUB_regions_4326.geojson** | Spatial reference file describing the study area sub-regions corresponding to parquet filename prefixes (i.e. VCI_basins.parquet) |
| **MetaData.pdf** | General information about the dataset content, formats, versioning, and input data sources. |
| **README.pdf** | Basin attribute descriptions and method references. |