# Peer review of "BCUB - A large sample ungauged basin attribute dataset for British Columbia, Canada."

_Earth System Science Data, 2023_

## Author Comment (AC1)

**Author's Thanks (2024-05-21)**: We are grateful for the time and care taken by the reviewer to consider our work and provide feedback. Some of the points raised made us think about the process and the data in new ways and highlight where important clarifications should be made. Please see below for our responses to specific feedback.

> **RC2 Comment:** *It would be nice if authors can convince us the necessity to have 1.2 million basins (sub-basins or sub-catchment), it's very difficult to find the one you are interested and very difficult to use all of them in a regional scale.*

**Author's Response**: The smallest monitored sub-basin operated by the BC Hydrometric Service (and likewise in the HYSETS dataset) is 1 km$^2$. In developing the BCUB dataset, we aimed to cover the range of basin sizes described by the set of monitored watersheds in BC since these are widely used in research and practice. The reason for this is that we want to comprehensively characterize the ungauged space, i.e. the set of ungauged basins above the size threshold. This will serve further research in what we might be missing with the current monitoring network. We will aim to highlight this goal somewhat more in the introduction.

We agree, the large number of sub-basins does create challenges for working with the dataset as a whole. We provide an example of working with (parts of) the data in a web-based Jupyter Notebook: ([https://dankovacek.github.io/bcub_demo/notebooks/7_Dataset_plot_example.html](https://dankovacek.github.io/bcub_demo/notebooks/7_Dataset_plot_example.html)) In an effort to support different use cases, we provide the data in two formats: i) a smaller and widely used (csv) format containing all sub-basin attributes with x,y coordinates of the sub-basin pour point and polygon centroids, and ii) the much larger basin polygon files in (geospatial) Parquet format (saved under basin_polygons in the data repository).

[Figure]

The Parquet format is supported by GDAL as of version 3.5, so QGIS must be compiled with GDAL >= 3.5 which is not default in some environments.
Please see the following for information about versions and compatibility:
[https://gis.stackexchange.com/questions/430973/importing-geoparquet-file-in-qgis](https://gis.stackexchange.com/questions/430973/importing-geoparquet-file-in-qgis)

Reading/writing Parquet in R:
https://arrow.apache.org/docs/r/reference/read_parquet.html

Reading/writing Parquet in Python:
https://arrow.apache.org/docs/python/parquet.html

Parquet is also implemented in Julia, MATLAB, Rust, Go, Java, C++, and others:
https://arrow.apache.org/docs/

> ***RC2 Comment***: *If possible, authors can introduce some specific implementations of these large-sample basins.*

Figures 9 and 10 in the manuscript give examples of specific questions that can be asked of this dataset. The ability to derive customized samples of basins by a wide range of characteristics may support experimental design, for example in temperature monitoring for quantifying the effect of land cover change on stream temperature. We are currently using the BCUB dataset as an input for a streamflow monitoring network optimization study. We will revise the manuscript to more fully describe these examples in the Usage Notes (section 3).

> ***RC2 Comment***: *It is also very difficult to find real observation of river discharge to support further analysis.*

**Author response**: The HYSETS dataset (Arsenault, 2019) provides streamflow observation at a large set of monitored locations along with their catchment polygons and attributes, whereas the BCUB dataset defines and describes attributes for sub-basins at all river network confluences and does not contain streamflow since the vast majority of confluences are ungauged. These two datasets can be combined to extrapolate information from monitored to unmonitored catchments. It should be noted that some work is required to map locations between HYSETS and BCUB datasets, and many historical monitoring locations no longer exist and their location coordinates were recorded with varying degrees of precision.

> ***RC2 Comment***: *Please check the unit of precipitation, e.g., 2028mm/day for gauge 1269663 must be wrong?*

**Author response**: Thank you for catching the typo, the precipitation index represents a mean annual value, which is derived from daily total precipitation from the DAYMET dataset. We have updated Table 2 to read "Mean Annual Precipitation" with the corresponding units [mm/year].

> ***RC2 Comment***: *What is the difference in the number of detected sub-basins when using the two spatial resolution?*

**Author response**: The number of sub-basins is a function of the minimum area threshold assumed in the stream network extraction process, which we set at 1 km$^2$ to match the smallest sub-basin in the streamflow monitoring network. Our hypothesis is that the number of basins is not expected to change significantly as a result of a change in the input DEM resolution. The key factors are:

1. The number of raster cells (pixels) representing the smallest sub-basin (1 km$^2$) increases by roughly an order of magnitude between the EarthENV (90m at the equator, ~$10^2$ pixels) and the USGS 3DEP (30m at the equator, ~$10^3$ pixels), and
2. The raster pixel dimension changes as a function of latitude, meaning the precision of one (integer) number of pixels increases with increasing latitude.

The smallest number of upstream accumulation cells in the BCUB dataset is 1507 which is a result of the projected grid dimension decreasing with increasing latitude (30m raster pixel would yield a minimum threshold of 1111 pixels). In the coarser resolution, this would represent 170 pixels owing to the factor of 9 area difference between the 30 and 90m grid dimensions. This number of pixels represents the worst case scenario where rounding to an integer number of pixels represents 0.6% rounding error. The BCUB dataset contains pour points at confluences, but not headwaters, unless they coincide, so only pour points (confluences) within 0.6% difference from the area threshold would be affected (included/excluded). Approximately 0.25% of the basins in the BCUB are less than 1.01km$^2$ representing a 1% deviation from the minimum basin area.

**References:**

1. Arsenault, R., Brissette, F., Martel, J.-L., Troin, M., Lévesque, G., Davidson-Chaput, J., Gonzalez, M. C., Ameli, A., and Poulin, A.: A comprehensive, multisource database for hydrometeorological modeling of 14,425 North American watersheds, Scientific Data, 7, 1–12, 2020.

---

## Author Comment (AC2)

**Author's Thanks (2024-05-21)**:   We are grateful for the time taken and the effort made by the reviewer to consider our work and provide feedback.  The points raised highlight important clarifications to both the content and delivery of the paper that undoubtedly improve its quality. Please see below for our responses to specific feedback.

> **RC1 Comment:** *A definition of the basin considered in this study is needed. Basin is a term that is interchangeable with catchment and watershed, but it typically refers to the entire drainage area of a river. In this article, 'basin' represents the local watershed of each river-reach. The term 'sub-catchment' or 'sub-basin' is more appropriate here.*

**Author's Response**:  As per your recommendation, we will add a definition of the term basin in our paper to clarify. While we agree that many of the basins considered in our dataset could be classified as sub-basins or sub-sub-basins, we use the term basin in a wider sense of the definition. This is in line with literature about ungauged basins. For example, the usage of "basin" in *"A decade of Prediction in Ungauged Basins (PUB)--a review"* (Hrachowitz et al., 2013) does not seem to refer exclusively to the entire drainage area of a river. To avoid confusion, we will explicitly define our use of "basin" at the start of Section 1.2.

> **RC1 Comment**: *To understand the process more easily, a flowchart showing different steps of BCUB database development in the methodology section would be helpful.*

**Author's Response**: We agree a diagram will provide a useful overview of the full process.  The diagram below will be added to the manuscript to represent the dataset development process:

[Figure]

*RC1 Comment:The reason for using the HydroBASINS watersheds (level-5 and 6) to subdivide the study region is understandable. However, the underlying hydrography data in the HydroBASINS and BCUB databases are different. So, there is a chance of missing a part of the sub-catchments located near the regional boundary in the BCUB database. For example, a part of the sub-catchment of the PCR region, located near the boundary between PCR and FRA, may overshoot to the FRA region due to hydrographic data inconsistencies. How was this issue addressed during the development of this database?*

**Author's Response**: Thank you for raising this important point. While the region polygons assembled from HydroBASINS are a helpful tool for organizing the data processing pipeline, indeed their use yields different bounds whose effect on sub-basin delineation is in the order of the size of the smallest sub-basins in the BCUB dataset.

The Caravan dataset (Kratzert et al., 2023) clearly describes the issue with aggregating attributes from catchment polygons that do not align with the HydroBASINS dataset. By independently deriving the region bounds from a single continuous DEM source (USGS 3DEP 30m grid), we avoid the problem of misalignment with HydroBASINS polygons, however it does not solve the problem of disagreement in region bounds defined independently of HydroBASINS.

Below is an outline of the process we use to independently redefine sub-region polygons from the DEM and quantify uncertainty in region bounds in the BCUB dataset.

[Figure]

The edge detail inset in the figure above shows an example segment of region boundaries aggregated from HydroBASINS (blue dashed line) compared to independently derived region

bounds.  The purple (Peace, PCR) and green (Fraser, FRA) coloured areas represent the region boundaries derived independently using the USGS 3DEP DEM (30m grid resolution), referred to here as the BCUB region boundaries.  White areas are gaps that remain following the iterative boundary definition process described below.  We define boundary deviations as polygons representing i) gaps between region bounds where the DEM resolution does not resolve which direction the small area drains, and ii) boundary overlaps when delineating from pour points in distinct basins with shared boundaries.

The process begins by applying a (5km) buffer to the region boundaries aggregated from level 5 and 6 HydroBASINS polygons, and using these buffered polygons as clipping masks on the DEM.  The purpose of this step is to avoid restricting the catchment boundary delineation by the clipping mask.  The covering set of polygons (catchments) are then delineated from the clipped DEM for each region, and the exterior edges (of the union of intersecting geometries) are checked to verify that they do not touch the edge of the buffered region polygon.  Where the edges intersect, we manually expand the buffer (DEM clipping mask) in QGIS and re-derive the covering set of catchments until the buffer is sufficient, i.e. the covering set of basins does not touch the edge of the clipping mask. The use of a buffer causes small catchments to be delineated which drain to basins in adjacent regions, and these are excluded by identifying that they are completely contained by the clipping mask of the neighbouring region.   The figure below illustrates the excluded vestigial edge sub-basins (purple) and the remaining covering basin set (orange).

[Figure]

Delineating region boundaries independently from the covering set of basins does not yield perfectly shared boundaries, but these deviations are substantially smaller than those resulting from aggregating the HydroBASINS levels 5 and 6 polygons. The distribution of the size of deviations from shared sub-region boundaries are shown in the figure below. The red series represents deviations between the BCUB region bounds and HydroBASINS-derived bounds (median area of 0.13 km$^2$), while the blue series represents deviations (overlaps and gaps) within the BCUB sub-region boundaries (median area = 0.03 km$^2$). Polygons smaller than 0.01 km$^2$, or 1% of the smallest sub-basin in the BCUB dataset were neglected.

[Figure]

We will incorporate a geometry flag attribute in the BCUB dataset for any sub-basin that intersects or touches at least one boundary deviation, and will include a decimal value to represent the total deviation area as a percentage of the sub-basin area. Where two different sub-basins claim the same area, either bordering sub-basin may overestimate the catchment area (indicated by a positive % value). Where an area is not claimed by any basin but is not necessarily endorheic, either bordering sub-basin may be underestimating the catchment area (indicated by a negative % value). The percentage represents the maximum expected percentage error from the uncertain boundary. The purpose of including these quantities is to communicate (some part of) the uncertainty in defining region bounds where the size of the uncertain area exceeds 1% of any sub-basin area. We will update the region boundaries in the data repository, and we will additionally provide the set of polygons representing boundary deviations as a .geojson file to facilitate corrections given updated information resolving these disagreements.

We additionally point out that a precise coastline definition (or ocean masking) at the resolution of the input DEM is important for the river network processing computation, otherwise vestigial river

segments occur in the ocean parallel to coastlines where the HydroBASINS polygons extend over ocean surface.  We crop the coastline using the NALCMS land cover data ocean pixels – the land cover data are well suited to the input DEM since the both products are provided in the same grid resolution.

Finally, these region boundary updates will require revising the BCUB dataset.  We will reprocess all affected sub-basins and update the dataset with the above additional information, namely the catchment delineation flag and the percent area represented by uncertain region boundaries.  The additional detail provided here will appear in some form in the manuscript. The code used to derive the region boundary deviations will be provided along with the existing validation code in the open-source code repository.  We believe these revisions will result in a more transparent and higher quality dataset, and we appreciate the reviewer raising this important detail.

> *RC1 Comment*: *It is sometimes difficult to follow the article due to inconsistencies in the statements. For example, the line 76 in the motivation section, "The accuracy of stream network delineation improves with increasing DEM resolution." The transition from the previous lines to this one is not smooth.*

**Author's Response**:
Agreed.  This point is made in a more appropriate context later in the text (lines 180-185) so we have removed the statement.

> *RC1 Comment*: *Another example of inconsistency is in line 134, where the delineation of the stream network is discussed after the description of the pour point selection process from the stream network. It would be more appropriate to discuss the stream network delineation process before selecting pour points.*

**Author's Response**:
Agreed.  The order of stream network extraction and pour point selection have been adjusted accordingly to improve the consistency overall narrative and sequencing of arguments.

> *RC1 Comment*: *Line 103: Please provide the minimum drainage area threshold used to delineate the stream network from USGS 3DEP*

**Author's Response**: The minimum drainage area threshold used is 1 km$^2$ which corresponds to the smallest sub-basin included in the HYSETS dataset (Arsenault et al. 2020) and to the smallest monitored basin in the British Columbia streamflow monitoring network. This reference is made explicit in the text, but your note identifies where (we agree) it should be placed earlier in the text. The text around line 103 has been updated to explicitly state the minimum threshold.

*RC1 Comment: Figure 7: This is a nice figure to show the impact of using DEM with different resolutions. The plot with colored density would be more helpful to understand the figure.*

**Author's Response**:

Figure 7 has been modified to show the distributions in both x and y, as shown below, which we hope adds clarity to the meaning of the figure. We tried a 2D (kernel) density plot to unsatisfactory effect. We believe the addition of x and y distributions are a reasonable compromise to sufficiently demonstrate the point that when increasing the input DEM resolution, the mean change in area is near zero while the corresponding change in perimeter is substantially greater than zero. We also add that the coefficient of determination ($R^2$) between x and y (area and perimeter deviation from baseline) is 0.00, which means that the marginal distributions of x and y do not lose any information relative to the joint distribution of x,y. We will modify the figure as shown below and add the coefficient of determination to the manuscript highlighting this point.

[Figure]

*RC1 Comment: When using QGIS version 3.28 to open the dataset, it displays the pour point location instead of the sub-basin polygon. Has the delineated sub-basin geometry been excluded from the database?*

**Author's Response**: The tabular file (BCUB_attributes_20240117.tab) contains the x,y coordinates of the pour point (ppt) and basin centroid ('centroid_x', 'centroid_y', 'ppt_lon_m_3005', 'ppt_lat_m_3005') while due to the very large file sizes, the polygon geometries are provided separately in the Parquet file format saved under the "basin_polygons" folder in the data repository:

[Figure]

Parquet is supported by GDAL as of version 3.5, so QGIS must be compiled with GDAL >= 3.5 which is not default in some environments.

Please see the following for information about versions and compatibility:
https://gis.stackexchange.com/questions/430973/importing-geoparquet-file-in-qgis

Reading/writing Parquet in R:
https://arrow.apache.org/docs/r/reference/read_parquet.html

Reading/writing Parquet in Python:
https://arrow.apache.org/docs/python/parquet.html

Parquet is also implemented in Julia, MATLAB, Rust, Go, Java, C++, and others:
https://arrow.apache.org/docs/

**References**:
1. Hrachowitz, Markus, et al. "A decade of Predictions in Ungauged Basins (PUB)—a review." Hydrological sciences journal 58.6 (2013): 1198-1255.
2. Arsenault, R., Brissette, F., Martel, J.-L., Troin, M., Lévesque, G., Davidson-Chaput, J., Gonzalez, M. C., Ameli, A., and Poulin, A.: "A comprehensive, multisource database for hydrometeorological modeling of 14,425 North American watersheds", Scientific Data, 7, 1-12, 2020.
3. Kratzert, Frederik, et al. "Caravan-A global community dataset for large-sample hydrology." Scientific Data 10.1 (2023): 61.

---

## Author Response (AR1)

**Authors' Response (Final)**

2024-07-02 - Daniel Kovacek & Steven Weijs

**Authors' thanks**: We would like to express our sincere gratitude for the time and effort of the editors and anonymous referees in reviewing and providing feedback on our work. The points raised by all highlight important clarifications, and the quality of the revised manuscript is significantly improved as a result.

Final responses to all referee comments to the draft manuscript are detailed below. The information is organized in the following order: i) referee comment, ii) author response, and iii) line numbers and/or sections identifying related manuscript changes.  Please note that page and line numbers referring to manuscript edits correspond to the **revised manuscript**.

**Responses to RC1 comments**

*RC1 Comment: A definition of the basin considered in this study is needed. Basin is a term that is interchangeable with catchment and watershed, but it typically refers to the entire drainage area of a river. In this article, 'basin' represents the local watershed of each river-reach. The term 'sub-catchment' or 'sub-basin' is more appropriate here.*

**Author's Response**:  While we agree that many of the basins considered in our dataset could be classified as sub-basins or sub-sub-basins, we use the term basin in a wider sense of the definition. This is in line with literature about ungauged basins. For example, the usage of "basin" in *"A decade of Prediction in Ungauged Basins (PUB)--a review"* (Hrachowitz et al., 2013) does not seem to refer exclusively to the entire drainage area of a river.  We agree there should be an explicit definition of our use of the term "basin", and it has been added at the start of Section 1.2.

**Corresponding Manuscript Edits**:

- An explicit definition of our usage of the term "basin" as "the local watershed of any confluence or outlet in a stream network." has been added at the start of Section 1.3 (line 85).
- All references to "basin" have been reviewed and changed to "sub-basin", "catchment", or "watershed" where appropriate to the specific context.

***RC1 Comment***: *To understand the process more easily, a flowchart showing different steps of BCUB database development in the methodology section would be helpful.*

**Author's Response**: We agree a diagram will provide a useful overview of the full process.

**Corresponding Manuscript Edits**:
- The diagram below has been added as Figure 2 to the manuscript (top of page 5) to represent the dataset development process:

[Figure]

***RC1 Comment***: *The reason for using the HydroBASINS watersheds (level-5 and 6) to subdivide the study region is understandable. However, the underlying hydrography data in the HydroBASINS and BCUB databases are different. So, there is a chance of missing a part of the sub-catchments located near the regional boundary in the BCUB database. For example, a part of the sub-catchment of the PCR region, located near the boundary between PCR and FRA, may overshoot to the FRA region due to hydrographic data inconsistencies. How was this issue addressed during the development of this database?*

**Author's Response**: Thank you for raising this important point. While the region polygons assembled from HydroBASINS are a helpful tool for organizing the data processing pipeline, indeed their use yields different bounds whose effect on sub-basin delineation is in the order of the size of the smallest sub-basins in the BCUB dataset.

The Caravan dataset (Kratzert et al., 2023) clearly describes the issue with aggregating attributes from catchment polygons that do not align with the HydroBASINS dataset. By independently deriving the region bounds from a single continuous DEM source (USGS 3DEP, 30m grid), we

avoid the problem of misalignment with HydroBASINS polygons, however it does not solve the problem of uncertainty in region bounds defined independently of HydroBASINS.

Below is an outline of the process we use to independently redefine sub-region polygons from the DEM and quantify uncertainty in region bounds in the BCUB dataset.

[Figure]

The edge detail inset in the figure above shows an example segment of region boundaries aggregated from HydroBASINS (blue dashed line) compared to independently derived region bounds. The purple (Peace, PCR) and green (Fraser, FRA) coloured areas represent the region boundaries derived independently using the USGS 3DEP DEM (30m grid resolution), referred to here as the BCUB region boundaries. White areas are gaps that remain following the iterative boundary definition process described below. We define boundary deviations as polygons representing i) gaps between region bounds where the DEM resolution does not resolve which direction the small area drains, and ii) boundary overlaps when delineating from pour points in distinct basins with shared boundaries.

The process begins by applying a (5km) buffer to the region boundaries aggregated from level 5 and 6 HydroBASINS polygons, and using these buffered polygons as clipping masks on the DEM. The purpose of this step is to avoid restricting the catchment boundary delineation by the clipping mask. The covering set of polygons (catchments) are then delineated from the clipped DEM for each region, and the exterior edges (of the union of intersecting geometries) are checked to verify that they do not touch the edge of the buffered region polygon. Where the edges intersect, we manually expand the buffer (DEM clipping mask) in QGIS and re-derive the covering set of catchments until the buffer is sufficient, i.e. the covering set of basins does not touch the edge of

the clipping mask. The use of a buffer causes small catchments to be delineated which drain to basins in adjacent regions, and these are excluded by identifying that they are completely contained by the clipping mask of the neighbouring region.  The figure below illustrates the excluded vestigial edge sub-basins (purple) and the remaining covering basin set (orange).

[Figure]

Delineating region boundaries independently from the covering set of basins does not yield perfectly shared boundaries, but these deviations are substantially smaller than those resulting from aggregating the HydroBASINS levels 5 and 6 polygons.  The distribution of the size of deviations from shared sub-region boundaries are shown in the figure below.  The red series represents deviations between the BCUB region bounds and HydroBASINS-derived bounds (median area of 0.13 km$^2$), while the blue series represents deviations (overlaps and gaps) within the BCUB sub-region boundaries (median area = 0.03 km$^2$).  Polygons smaller than 0.01 km$^2$, or 1% of the smallest sub-basin in the BCUB dataset were neglected.

We will incorporate a geometry flag attribute in the BCUB dataset for any sub-basin that intersects or touches at least one boundary deviation, and will include a decimal value to represent the total deviation area as a percentage of the sub-basin area.  Where two different sub-basins claim the same area, either bordering sub-basin may overestimate the catchment area (indicated by a positive % value).  Where an area is not claimed by any basin but is not necessarily endorheic, either bordering sub-basin may be underestimating the catchment area (indicated by a negative % value).  The percentage represents the maximum expected percentage error from the uncertain boundary.  The purpose of including these quantities is to communicate (some part of) the uncertainty in defining region bounds where the size of the uncertain area exceeds 1% of any sub-basin area.  We will update the region boundaries in the data repository, and we will

additionally provide the set of polygons representing boundary deviations as a .geojson file to facilitate corrections given updated information resolving these disagreements.

We additionally point out that a precise coastline definition (or ocean masking) at the resolution of the input DEM is important for the river network processing computation, otherwise vestigial river segments occur in the ocean parallel to coastlines where the HydroBASINS polygons extend over ocean surface. We crop the coastline using the NALCMS land cover data ocean pixels – the land cover data are well suited to the input DEM since the both products are provided in the same grid resolution.

Finally, these region boundary updates will require revising the BCUB dataset. We will reprocess all affected sub-basins and update the dataset with the above additional information, namely the catchment delineation flag and the percent area represented by uncertain region boundaries. The additional detail provided here will appear in some form in the manuscript. The code used to derive the region boundary deviations will be provided along with the existing validation code in the open-source code repository. We believe these revisions will result in a more transparent and higher quality dataset, and we appreciate the reviewer raising this important detail.

**Corresponding Manuscript Edits**:

- Section 2.2.1 (page 10-11) has been added to describe the treatment of uncertain sub-region boundaries.
- Figure 5 has been added at the top of page 11 to illustrate the problem of uncertain region bounds,

- Figure 6 has been added at the top of page 11 to quantify the effect of the treatment described in section 2.2.1 on reducing the size of boundary deviations.
- Three additional columns have been added to the dataset to (1) flag where uncertain boundaries border with catchments in the BCUB and to quantify the uncertain area (2-gap, 3-overlap) as a percentage of the catchment area.  Definitions of these attributes have been added to subsection 2.1.2 at lines 161 to 166.
- Table 3 has been updated to reflect the additional metadata attributes (top of page 22).

*RC1 Comment: It is sometimes difficult to follow the article due to inconsistencies in the statements. For example, the line 76 in the motivation section, "The accuracy of stream network delineation improves with increasing DEM resolution." The transition from the previous lines to this one is not smooth.*

**Author's Response**:
We agree.

**Corresponding manuscript edits**:

- The point about accuracy of stream networks was moved to Section 2.2 (line 190) where it is more relevant.

*RC1 Comment: Another example of inconsistency is in line 134, where the delineation of the stream network is discussed after the description of the pour point selection process from the stream network. It would be more appropriate to discuss the stream network delineation process before selecting pour points.*

**Author's Response**:
Agreed.  The order of stream network extraction and pour point selection have been adjusted accordingly to improve the consistency overall narrative and sequencing of arguments.

**Corresponding manuscript edits**:

- A point-by-point ordered summary of the data collection and processing has been added to the introduction of section 2.1 (page 5, lines 112-131). The more detailed information about points 4-6 (lines 123-130) have been reordered in subsections 2.1.1 (line 132), 2.1.2 (line 153), and 2.1.3 (line 167) to correspond with the sequence in which they are introduced.
- Small edits to have been made throughout the manuscript to improve the overall grammar and organization with deliberate care to preserve the content, meaning, and interpretation of the arguments.

*RC1 Comment: Line 103: Please provide the minimum drainage area threshold used to delineate the stream network from USGS 3DEP*

**Author's Response**: The minimum drainage area threshold used is 1 km$^2$ which corresponds to the smallest sub-basin included in the HYSETS dataset (Arsenault et al. 2020) and to the smallest monitored basin in the British Columbia streamflow monitoring network. This reference is made explicit in the text, but your note identifies where (we agree) it should be placed earlier in the text.

**Corresponding manuscript edits**:

- We have moved the explicit reference to minimum drainage area threshold earlier in the text as recommended to line 109.

*RC1 Comment: Figure 7: This is a nice figure to show the impact of using DEM with different resolutions. The plot with colored density would be more helpful to understand the figure.*

**Author's Response**:
Figure 7 (Figure 10 in the revised manuscript) has been modified (see below) to show the probability densities in both x and y, which we believe adds clarity to the meaning of the figure. We tried a 2D (kernel) density plot to unsatisfactory effect due to either requiring a less interpretable colour mapping or x and y units. We believe the addition of probability densities of x and y are a reasonable compromise to effectively communicate the point that when increasing the input DEM resolution, the mean change in area is near zero while the corresponding change in perimeter is substantially greater than zero. We also add here that the coefficient of determination (R$^2$) between x and y (area and perimeter deviation from baseline) is zero.

[Figure]

**Corresponding manuscript edits**:

- Figure 7 in the draft manuscript is now Figure 10 (shown above), located at the top of page 15.

***RC1 Comment:*** *When using QGIS version 3.28 to open the dataset, it displays the pour point location instead of the sub-basin polygon. Has the delineated sub-basin geometry been excluded from the database?*

**Author's Response**:  The tabular file (BCUB_attributes_20240117.tab) contains the x,y coordinates of the pour point (ppt) and basin centroid ('centroid_x', 'centroid_y', 'ppt_lon_m_3005', 'ppt_lat_m_3005') while due to the very large file sizes, the polygon geometries are provided separately in the Parquet file format saved under the "basin_polygons" folder in the data repository:

[Figure]

To avoid issues with limited memory (<64GB), the default geometry in the parquet files is set to the pour point to use considerably less memory than the polygon geometries.  We recommend filtering geometries based on specific questions before loading polygons for visualization.

Parquet is supported by GDAL as of version 3.5, so QGIS must be compiled with GDAL >= 3.5 which is not default in some environments.

Please see the following for information about versions and compatibility:
https://gis.stackexchange.com/questions/430973/importing-geoparquet-file-in-qgis

Reading/writing Parquet in R:
https://arrow.apache.org/docs/r/reference/read_parquet.html

Reading/writing Parquet in Python:
https://arrow.apache.org/docs/python/parquet.html

Parquet is also implemented in Julia, MATLAB, Rust, Go, Java, C++, and others:
https://arrow.apache.org/docs/

**Corresponding manuscript edits**:
- Suggested resources for working with the data have been added in lines 312-317.
- A reference pointing to the notebook tutorial demonstrating import and use of data from the Parquet format has been added at line 331.

**Responses to RC2 comments**

*RC2 Comment:* *It would be nice if authors can convince us the necessity to have 1.2 million basins (sub-basins or sub-catchment), it's very difficult to find the one you are interested and very difficult to use all of them in a regional scale.*

**Author's Response**:   The smallest monitored sub-basin operated by the BC Hydrometric Service (and likewise in the HYSETS dataset) is 1 km$^2$.  In developing the BCUB dataset, we aimed to cover the range of basin sizes described by the set of monitored watersheds in BC since these are widely used in research and practice.  The reason for this is that we want to comprehensively characterize the ungauged space, i.e. the set of ungauged basins above the size threshold. This will serve further research in what we might be missing with the current monitoring network. We will aim to highlight this goal somewhat more in the introduction.

We agree that the large number of sub-basins does create challenges for working with the dataset as a whole.  We provide an example of working with (parts of) the data in a web-based Jupyter Notebook: (https://dankovacek.github.io/bcub_demo/notebooks/7_Dataset_plot_example.html) In an effort to support different use cases, we provide the data in two formats: i) a smaller and widely used (csv) format containing all sub-basin attributes with x,y coordinates of the sub-basin pour point and polygon centroids, and ii) the much larger basin polygon files in (geospatial) Parquet format (saved under basin_polygons in the data repository).

[Figure]

The Parquet format is supported by GDAL as of version 3.5, so QGIS must be compiled with GDAL >= 3.5 which is not default in some environments.

Please see the following for information about versions and compatibility: https://gis.stackexchange.com/questions/430973/importing-geoparquet-file-in-qgis

Reading/writing Parquet in R: https://arrow.apache.org/docs/r/reference/read_parquet.html

Reading/writing Parquet in Python:

https://arrow.apache.org/docs/python/parquet.html

Parquet is also implemented in Julia, MATLAB, Rust, Go, Java, C++, and others:
https://arrow.apache.org/docs/

**Corresponding manuscript edits**:

- The aim of covering the range of catchment areas represented in large sample hydrology is described in lines 45-49. The number of basins in the BCUB is a function of aiming to cover the range of spatial scales described by related datasets.

*RC2 Comment*: *If possible, authors can introduce some specific implementations of these large-sample basins.*

Figures 9 and 10 in the manuscript give examples of specific questions that can be asked of this dataset. The ability to derive customized samples of basins by a wide range of characteristics may support experimental design, for example in temperature monitoring for quantifying the effect of land cover change on stream temperature. We are currently using the BCUB dataset as an input for a streamflow monitoring network optimization study.

**Corresponding manuscript edits**:

- Suggested resources for working with the data are provided in lines 312-316.
- A link to and description of the notebook (tutorial) demonstrating import and use of data from the Parquet format has been added in lines 329-332.

*RC2 Comment*: *It is also very difficult to find real observation of river discharge to support further analysis.*

**Author response**: The HYSETS dataset (Arsenault, 2019) provides streamflow observation at a large set of monitored locations along with their catchment polygons and attributes, whereas the BCUB dataset defines and describes attributes for sub-basins at all river network confluences and does not contain streamflow since the vast majority of confluences are ungauged. These two datasets can be combined to extrapolate information from monitored to unmonitored catchments. It should be noted that some work is required to map locations between HYSETS and BCUB datasets, and many historical monitoring locations no longer exist and their location coordinates were recorded with varying degrees of precision.

*RC2 Comment*: *Please check the unit of precipitation, e.g., 2028mm/day for gauge 1269663 must be wrong?*

**Author response**: Thank you for catching the typo, the precipitation index represents a mean annual value, which is derived from daily total precipitation from the DAYMET dataset. We have updated Table 2 to read "Mean Annual Precipitation" with the corresponding units [mm/year].

**Corresponding manuscript edits**:

- Table 2 (page 21) has been updated with the correct units (Annual precipitation, mm/year)

*RC2 Comment:* *What is the difference in the number of detected sub-basins when using the two spatial resolution?*

**Author response**: The number of sub-basins is a function of the minimum area threshold assumed in the stream network extraction process, which we set at 1 km$^2$ to match the smallest catchment in the streamflow monitoring network. Our hypothesis is that the number of basins is not expected to change significantly as a result of a change in the input DEM resolution. The key factors to consider are:
1. The processing of DEM and flow direction raster data to define the stream network assumes a minimum area threshold.
2. The number of raster cells (pixels) representing the smallest sub-basin (1 km$^2$) increases by roughly an order of magnitude between the EarthENV (90m at the equator, ~$10^2$ pixels) and the USGS 3DEP (30m at the equator, ~$10^3$ pixels), and
3. The raster pixel dimension changes as a function of latitude, meaning the precision of one (integer) number of pixels increases with increasing latitude.

The smallest number of upstream accumulation cells in the BCUB dataset is 1507 which is a result of the projected grid dimension decreasing with increasing latitude (30m raster pixel would yield a minimum threshold of 1111 pixels). In the coarser resolution, this would represent 170 pixels owing to the factor of 9 area difference between the 30 and 90m grid dimensions. This number of pixels represents the worst case scenario where rounding to an integer number of pixels represents 0.6% rounding error. The BCUB dataset excludes vestigial headwater points, so only pour points (confluences) within 0.6% difference from the area threshold should be affected (included/excluded) between resolutions. Approximately 0.25% of the basins in the BCUB are less than 1.01km$^2$ representing a 1% deviation from the minimum basin area.

**Corresponding manuscript edits**:

- The effects of changing resolution are highlighted in sections 2.2.2 and 2.2.3, in particular in figures 7 to 10.
- We did not derive the full attribute set from the lower resolution DEM but the full replication code is provided which can be used for comparisons between datasets and methods.

**References**:

1.  Hrachowitz, Markus, et al. "A decade of Predictions in Ungauged Basins (PUB)—a review." Hydrological sciences journal 58.6 (2013): 1198-1255.
2.  Arsenault, R., Brissette, F., Martel, J.-L., Troin, M., Lévesque, G., Davidson-Chaput, J., Gonzalez, M. C., Ameli, A., and Poulin, A.: "A comprehensive, multisource database for hydrometeorological modeling of 14,425 North American watersheds", Scientific Data, 7, 1-12, 2020.
3.  Kratzert, Frederik, et al. "Caravan-A global community dataset for large-sample hydrology." Scientific Data 10.1 (2023): 61.

---

## Author Response (AR2)

**Author's Response (Revision Round 2)**

2024-10-09 - Daniel Kovacek & Steven Weijs

**Authors' thanks**: We are grateful to the editors and anonymous reviewers for their time dedicated to reviewing and providing feedback on our manuscript. The points raised have meaningfully improved the quality of the final revision.

A brief response to the editor's (additional) comments regarding figures is presented first below. No comments were provided by Reviewer #2, so we include final responses to comments from Reviewer #3 only. The information is organized in the following order: i) reviewer comment, ii) author response, and iii) line numbers and/or sections identifying related manuscript changes. Please note that page and line numbers referring to manuscript edits correspond to the **revised manuscript**.

**Responses to Editor's comments**

*Editor's Comment: I find that the quality of the figures in the paper is still not satisfactory for publication in ESSD. For instance, Figures 4 and 5 are particularly difficult to follow..*

**Author's Response**: We have revised the maps (Figures 1, 3, 4, 5, and 9) to be more consistent in appearance and have added geographic reference features to maps where appropriate. Where gridlines are presented, a North arrow is left out since this information is indicated by the latitude and longitude grid line labels. Thank you for being persistent in the feedback regarding figure quality, we hope these revisions are in line with the standard expected from the journal. In addition, care has been taken to ensure that axis and legend labels in quantitative figures are larger and more consistent in their appearance. Quantitative figures (6, 7, 8, 11, and 12) have been updated to increase font sizes overall and use a consistent font where possible. Note that Figure 8 had to be reprocessed and it drew a different random sample of sub-basins so the distribution has changed slightly but the conclusion about the effect of DEM resolution on the computed slope does not change.

**Responses to RC3 comments**

*RC3 Comment: "From Table 1, it is not clear if these are the only provided fields because Section 2.1.3 addes some new fields. … It will also be useful to provide a new table (similar to what's provided in README) summarizing the final released data, fields, units, etc."*

**Author's Response**:  Table 1 represents a summary of **data sources** used to derive all attributes. Table 2 (referenced at lines 170 and 200) contains all the attributes, descriptions, units, etc as you suggest.   Unfortunately due to the size of the table owing to the number of attributes / fields, it is relegated to the end of the paper following the references automatically by LaTeX.  A reference to Table 2 has been added nearer to Table 1 for improved clarity.

**Corresponding Manuscript Edits**:
- The introduction of section 2.1 (near line 105) has been revised as follows: *"Table 1 provides a summary of the geospatial data sources—including digital elevation, land cover, soil, and climate datasets—that attributes of ungauged basins were extracted from. These datasets were processed through the data preparation pipeline outlined in Figure 2, and the resulting attributes are listed and described in Table 2."*

*RC3 Comment*: *…many recently published LSH work in ESSD already provides more fields than what's provided here for British Columbia. Can the authors provide more fields, or justifications on why they are not provided here, by comparing it with existing work?*

**Author's Response**: Since the BCUB dataset does not include streamflow, it doesn't strictly belong with LSH datasets. The dataset was initially created to provide a basis for comparing streamflow monitoring networks to the much larger ungauged areas using similar attributes, which we believe offers an important comparison and a novel one. The primary goal, in our case, is optimizing streamflow monitoring networks—a task, to our knowledge, not previously approached by characterizing the ungauged space to a level of detail approaching that which monitored catchments are described in the LSH literature.

Basin delineation is a source of uncertainty in the estimation of catchment attributes, and this is one reason other datasets, like the Caravan dataset (Kratzert et al. 2023), initially excluded catchments smaller than 100 km² though we not only in very recent months this constraint has been revised in part. However, as noted in lines 78-81 of the manuscript regarding the HYSETS dataset (Arsenault et al. 2020), which is a major component of the Caravan dataset, catchments under 50 km² account for nearly one-third of monitored catchments in British Columbia. We believe this exclusion introduces a significant bias but also presents a meaningful opportunity for further exploration which our dataset aims to support.

A major component of this work was the delineation of catchment boundaries for a very large set of ungauged catchments.  LSH datasets typically use catchment polygons provided by official governing bodies, in HYSETS (which is a component of Caravan) the catchment polygons are from the Water Survey of Canada (WSC), the U.S. Geological Survey (USGS), and the (Mexican) Comisión Nacional del Agua (Conagua).  The importance of deriving catchment polygons from consistent sources is a component of the design criteria of the dataset (line 50-54), and some consequences of not doing so are described in the Technical validation sections, 2.2.2 and 2.2.3, namely inconsistency of certain terrain attributes when using elevation datasets at different

resolutions. The emphasis placed on catchment delineation is a more direct treatment of one source of uncertainty in catchment attributes that is not often addressed in the LSH literature.

We recognize the set of attributes we provide in the BCUB is not as comprehensive or numerous as other datasets. Given the rapid development of attributes in the LSH literature in recent years, we did not aim to provide the most complete and current set of attributes as this is a moving target. In addition, the utility and uncertainty in data sources underlying certain attributes (i.e. soil, geology) have been noted in the literature (Beck et al. 2015; Addor et al. 2018). Given that our dataset is roughly two orders of magnitude larger than the largest LSH dataset (Caravan), processing attributes is a large undertaking, but our paper provides a data processing pipeline template to support continuous development of customizable attribute sets.

As far as the number of catchment attributes, our goal was to provide an initial set that is representative of the dominant groups of attributes appearing in the literature, namely terrain, land cover, climate, and soil, and to provide complete code and tutorial-like instructions to offer a highly extensible and transparent data product. We believe this approach supports dataset extension and customization for specific needs across disciplines in light of the accelerating development of remote sensing data products. By providing the full code in addition to accompanying tutorials, we believe this dataset sets a precedent as far as being explicit about what attributes represent and how they are derived.

**Corresponding Manuscript Edits**:
- Lines 34-35 in the introduction have been revised to clarify the point about ungauged dataset as a basis of comparison for monitoring networks.
- A clearer gap statement between LSH datasets and hydrographic datasets has been added to the end of Section 1.1 to describe the need for the BCUB.
- Section 1.2 has been subdivided into hydrographic datasets (1.2.1) and LSH datasets (1.2.2). The new section 1.2.2 includes a concise description of key points in the evolution of LSH.
- A paragraph was added at the end of section 1.3 as follows: *"Our goal with the BCUB dataset was to provide a representative set of catchment attributes that cover key groups commonly found in the literature—terrain, land cover, climate, and soil. While our attribute set is not as extensive as those found in the LSH literature, we prioritized creating a transparent, extensible data product with complete code and tutorial-like information. Given the rapid development of attributes in LSH research, we focused on providing a solid framework rather than the most exhaustive or up-to-date set of attributes."*

*RC3 Comment*: *Why are streamflow not served here?*

**Author's Response**: The British Columbia Ungauged Basin dataset focuses on the much larger set of catchments that are unmonitored. The novelty of this dataset is in providing catchment attributes similar to those found in the LSH literature but for a much larger set of catchments where streamflow measurements have not been collected.

**Corresponding Manuscript Edits**:
- Splitting section 1.2 into two sub-sections (as described above) should hopefully make it more clear how this dataset is positioned in the gap statement, namely the closing statement of section 1.1 (~ line 60).

*RC3 Comment: The current introduction is lacking a comprehensive background on the state-of-the-art knowledge on other existing global-scale or continental-scale LSH datasets. For example, some of the cited literature in Line 12 is only the geospatial datasets but not the LSH datasets made available. I suggest the authors to do a more comprehensive literature summary, and place the BC work into better context of the community development.?*

**Author's Response**:
We agree that a clearer link between the availability of geospatial datasets and the emergence of LSH datasets should be made in the introduction. We focused the description of state of the art datasets on comparable hydrographic datasets for brevity, however since this dataset is intended to represent a bridge between the two, it is important to incorporate the LSH literature as the reviewer points out.

**Corresponding Manuscript Edits**:
- The first paragraph of the introduction has been changed to explicitly describe the link between geospatial (hydrographic) and LSH datasets.

**References**:
1. Arsenault, R., Brissette, F., Martel, J.-L., Troin, M., Lévesque, G., Davidson-Chaput, J., Gonzalez, M. C., Ameli, A., and Poulin, A.: "A comprehensive, multisource database for hydrometeorological modeling of 14,425 North American watersheds", Scientific Data, 7, 1-12, 2020.
2. Kratzert, F., Nearing, G., Addor, N., Erickson, T., Gauch, M., Gilon, O., Gudmundsson, L., Hassidim, A., Klotz, D., Nevo, S., et al.: Caravan-A global community dataset for large-sample hydrology, Scientific Data, 10, 61, 2023.
3. Beck, H. E., De Roo, A., and van Dijk, A. I.: Global maps of streamflow characteristics based on observations from several thousand catchments, Journal of Hydrometeorology, 16, 1478–1501, 2015.
4. Addor, N., Nearing, G., Prieto, C., Newman, A., Le Vine, N., and Clark, M. P.: A ranking of hydrological signatures based on their predictability in space, Water Resources Research, 54, 8792–8812, 2018.395
5. Addor, N., Do, H. X., Alvarez-Garreton, C., Coxon, G., Fowler, K., & Mendoza, P. A. (2019). Large-sample hydrology: recent progress, guidelines for new datasets and grand challenges. Hydrological Sciences Journal, 65(5), 712–725. https://doi.org/10.1080/02626667.2019.1683182

---

## Author Response (AR3)

**Author's Response (Revision Round 3)**

2024-11-07 - Daniel Kovacek & Steven Weijs

**Authors' thanks**: We are grateful to the editors for their time dedicated to reviewing and providing feedback on our manuscript.  The persistence in seeking a higher standard is appreciated.

The response information below is organized in the following order: i) editor comment, ii) author response, and iii) line numbers and/or sections identifying related manuscript revisions with description of changes.  Please note that page and line numbers referring to manuscript edits correspond to the **revised manuscript**.

**A note regarding the track-changes file**: we removed text-wrapping on figures to prevent these figures and their captions from being cut off.

**Responses to Editor's comments**

*Editor's Comment: I think the BCUB you present belongs to large sample hydrology (LSH) datasets, even though it does not include hydrological, climatic, and physical characteristics. In the revised manuscript, you defined BCUB as "hydrographic datasets," which makes the concepts complex and not easy to follow. I suggest it may be better to illustrate the contribution of BCUB within the concept framework of LSH.*

**Author's Response**:   The the revisions in question were partially in response to a comment from one of the reviewers ("RC3: *Why are streamflow not served here?*"), which suggested it was not clear enough that the dataset is not an LSH dataset due to its lack of streamflow data.  Revising the introduction was also done to clarify the gap statement, and we use the examples of LSH and hydrography to establish the points between which the BCUB is positioned.   We feel including a description of hydrographic datasets is an important point of contrast, though we also see the editor's point regarding positioning the BCUB dataset more in the concept framework of LSH.

**Corresponding Manuscript Edits (related edits from last revision)**:
- Section 1.1 was revised to emphasize the BCUB as a complement to LSH.

*Editor's Comment:  For Figure 1, the green color is used to show both catchments and the active monitoring network, making the figure difficult to follow.*

**Author's Response**:  Agreed, the contrast can be improved.

**Corresponding Manuscript Edits**
- Figure 1 has been updated as follows: BCUB region changed from green to grey, symbol sizes in inset map and legend increased for better visibility.

*Editor's Comment: Figure 2 is not completely displayed, and the figure caption is missing.*

**Author's Response**:  The automatic formatting in the version with tracked changes "Latex diff" shifted Figure 2 out of frame and unfortunately there was not an obvious fix.  For the current tracked changes file we removed text wrapping to avoid this issue.  Neither the figure nor the caption changed since the previous revision, and they are rendered "correctly" in the revised manuscript.

**Corresponding Manuscript Edits**
- No changes were made. Latexdiff rendering was edited to prevent images and captions from being shifted off the page.

*Editor's Comment:In Figure 3, explain "NA" in the figure caption. Add (a) and (b) for the two subplots, and include a scale bar for Figure 3a. The red color is used to show both NA Level 5 basins and the BC border, making the figure difficult to follow.*

**Author's Response**:  The N.A. stands for North America and we agree it should be defined in the manuscript.  Same comment regarding the tracked changes file.

**Corresponding Manuscript Edits**
- Figure 3 has been updated to combine the components of the previous revision into a single figure.  This way the figure can be rendered much larger for clarity without losing the intent of the figure to demonstrate how the sub-regions were derived.  The legend has been changed to N.Am. and the full expansion "North America" has been added in the caption. We verified that BC is defined as British Columbia earlier in the manuscript (introduction).

*Editor's Comment:  Regarding Figure 4, is there a lake outflow point? Streams in the lakes are not clear.*

**Author's Response**:  The "Lake Inflows" label for the outflow point is indeed misleading.  Since lake outlets are not distinguished from inlets explicitly in the dataset, we did not add a unique layer to the figure.  We considered several more general terms and decided on the more general labels "Lake Inflow/Outflow" in figure legend and also refer to river-lake connections in the caption.

**Corresponding Manuscript Edits**

- The yellow triangle symbol has been renamed to "Lake Inflow/Outflow".  The transparency of the lake layer is increased slightly to make the underlying vestigial stream lines more visible without adding too much visual clutter.

*Editor's Comment:  For Figure 5, different datasets generate slightly variable basin boundaries. This is a straightforward point. Is there any additional information we can gather from this figure? If not, I suggest deleting this figure. The differences in basin boundaries can be demonstrated in Figure 10.*

**Author's Response**:  We agree, the same information is clearly illustrated in Figure 10..

**Corresponding Manuscript Edits**
- Figure 5 has been removed along with the paragraph explaining it.

*Editor's Comment: In Figure 6, briefly explain in the figure caption what information we can gather from the figure. The font in axis labels and the legend is too small.*

**Author's Response**:  While some boundary uncertainty remains, the revised method reduces the median uncertainty for the smallest catchments from over 10% to about 2.5%, addressing the initial concern raised by the reviewer.

**Corresponding Manuscript Edits**
- The caption of Figure 6 (now Figure 5) has been revised as follows:  "Boundary uncertainties are significantly reduced relative to the smallest catchments in the dataset (1 $km^2$) when region bounds are generated from the same DEM source as the catchments (blue, median uncertainty 0.025 $km^2$), compared to HydroBASINS-derived regions (red, median uncertainty 0.13 $km^2$).  Axes and legend labels have been reformatted to use larger fonts.

*Editor's Comment: Figures 7 and 8 should be combined.*

- **Author's Response / Manuscript edits**:  We agree, Figures 7 and 8 and their captions have been combined into one figure (now Figure 6).  Please note that in order to make the tracked-changes file not get cut off from the page, we had to insert a page break, and the deletion of old text makes the images not display side-by-side.  The final rendering can be seen clearly in the revised manuscript, and we trust this is a fair compromise to ensure that all figures and the corresponding changes are rendered on the page and not cut off.

***Editor's Comment:***  *The caption for Figure 9 is incomplete, and this figure is difficult to follow. More explanation should be added to the figure caption. For example, what information can we learn from area and perimeter deviations?*

**Author's Response / Manuscript Edit**:  We have removed the figure as it is perhaps unnecessarily overcomplicating a simple result.  The description of the result has been reviewed for clarity in Section 2.2.3.

***Editor's Comment:*** *Figures 11 and 12 should be combined. Add scale bars.*

**Author's Response**:  Please see the next comment response.

***Editor's Comment:***  *Currently, there are no figures showing the overall pattern of the BCUB dataset. I suggest Figures 11 and 12 be plotted much larger to more clearly show at least a part of the BCUB dataset map. Otherwise, Figure 10 should be modified, or a new figure should be added for this purpose. At present, most figures are about technical details of producing the dataset rather than the dataset itself. Particularly, the rivers and lakes that you mentioned are not shown throughout the paper. Readers need to see how the BCUB dataset map looks beyond some technical detail figures.*

**Author's Response**: We agree, the figures can be given more space to better illustrate examples of what the dataset can be used for.  We have plotted Figures 11 and 12 each to full page width per your suggestion.  We modified the second example figure to represent a question asked of the full dataset.  This is a dataset representing catchment-level attributes, so queries, and visualizations thereof, will be catchment-based.

**Corresponding Manuscript Edits**
- Figures 11 and 12 (now 8 and 9) have been modified to full page width.

**References**:

1. Mandelbrot, B.: How long is the coast of Britain? Statistical self-similarity and fractional dimension, science, 156, 636–638, 1967.

---

## Author Response (AR4)

**Author's Response (Revision Round 4)**

2024-11-22 - Daniel Kovacek & Steven Weijs

**Authors' thanks**: Thank you once again for your diligence in reviewing and providing feedback on our manuscript.

**Responses to Editor's comments**

*Editor's Comment: Figure 1 does not appear to be updated as described in the response letter ("Figure 1 has been updated as follows: BCUB region changed from green to grey, symbol sizes in inset map and legend increased for better visibility").*

**Author's Response**:   I was not able to render both previous and updated versions of figures in the tracked changes document.

**Corresponding Manuscript Edits (related edits from last revision)**:
- No revisions made to the figure, please refer to the revised manuscript provided to see the updated version of Figure 1.

*Editor's Comment:  In Figure 5, the yellow triangle symbol is still labeled "Lake Inflows" instead of "Lake Inflow/Outflow". Additionally, the stream confluence points (green circles) seem to overlap. Is there a reason for two confluence points being placed at each confluence?.*

**Author's Response**:  Please note that the figure in question is now Figure 4.   The definition of green circles was defined explicitly in line 185 and in the caption of Figure 4.  Both of these have been revised to state that the green circles represent pour points that define catchments at confluences, including upstream branches and their combination.

**Corresponding Manuscript Edits**
- The representation of green circles is defined in line 184 as "Green circles represent pour points of catchments defining each upstream branch of a confluence plus their combination."  This clarification is also made in the caption of Figure 4.  The legend has been updated in Figure 4 to label green circles as "catchment pour point" and the red x label as "spurious pour point".  In the legend label it is also clarified that yellow triangles are also pour points defining catchment outlets. A definition of "pour point" has been added.

---

## Author Response (AR5)

**Author's Response (Final)**

2024-11-28 - Daniel Kovacek & Steven Weijs

**Authors' thanks**: A final thanks to the editor for your attention and effort to improve the quality of this work..

A revision to the manuscript was uploaded to include an expression of gratitude to a fellow researcher for their correspondence related to their work. Minor changes were made at the end of the introduction (lines 28-42) to improve grammar.